# Activity of MukBEF for chromosome management in *E. coli* and its inhibition by MatP

**Mohammed Seba, Frederic Boccard, Stéphane Duigou***

Université Paris-Saclay, CEA, CNRS, Institute for Integrative Biology of the Cell, Gif-sur-Yvette, France

**Abstract** Structural maintenance of chromosomes (SMC) complexes share conserved structures and serve a common role in maintaining chromosome architecture. In the bacterium *Escherichia coli*, the SMC complex MukBEF is necessary for rapid growth and the accurate segregation and positioning of the chromosome, although the specific molecular mechanisms involved are still unknown. Here, we used a number of in vivo assays to reveal how MukBEF controls chromosome conformation and how the MatP/*matS* system prevents MukBEF activity. Our results indicate that the loading of MukBEF occurs preferentially on newly replicated DNA, at multiple loci on the chromosome where it can promote long-range contacts in cis even though MukBEF can promote long-range contacts in the absence of replication. Using Hi-C and ChIP-seq analyses in strains with rearranged chromosomes, the prevention of MukBEF activity increases with the number of *matS* sites and this effect likely results from the unloading of MukBEF by MatP. Altogether, our results reveal how MukBEF operates to control chromosome folding and segregation in *E. coli*.

**\*For correspondence:**
stephane.duigou@i2bc.paris-saclay.fr

**Competing interest:** The authors declare that no competing interests exist.

## eLife assessment

This **important** work combines DNA contact analysis and controlled genome rearrangements to investigate the processes that organize the *E. coli* chromosome, with a particular focus on how the SMC-related complex MukBEF is regulated. The evidence supporting the conclusions is **compelling**, with time-resolved experiments and analysis of mutant strains. The work will be of broad interest to chromosome biologists and bacterial cell biologists.

## Introduction

Structural maintenance of chromosomes (SMC) complexes play key roles in many processes involved in chromosome management, from genome maintenance, interphase chromatin organization, sister chromatids alignment, chromosome folding and condensing, to DNA recombination at specific stages of the cell cycle (*Yatskevich et al., 2019*). A general model for SMC complex activity relies on their properties to bridge DNA elements and by doing so build DNA loops in cis and hold together the sister chromatids in trans; to do that, they bind DNA and processively extrude a DNA loop in an ATPase-driven loop extrusion, thereby compacting and organizing DNA (*Davidson et al., 2019*; *Ganji et al., 2018*; *Kong et al., 2020*). Yet many molecular features that determine the activity of the various SMC complexes are still unclear and it is not known whether they work using the same basal mechanisms (*Bürmann et al., 2021*; *Davidson et al., 2019*; *Pradhan et al., 2022*). For example, it is not yet clear whether (i) the activity of SMC complexes is mediated by a single complex or if it involves cooperation between several complexes that organize into dimers or even oligomers (*Badrinarayanan et al., 2012b*; *Hassler et al., 2018*), (ii) the DNA extrusion involves a topological or

nontopological mechanism, i.e., does the DNA pass through the SMC ring topologically (*Bürmann et al., 2021*; *Davidson et al., 2019*; *Pradhan et al., 2022*), (iii) the activity of SMC complexes relies on the folding capacities of the SMC coiled-coil arms facilitating large-scale conformational changes (*Bürmann et al., 2019*), and (iv) multiple complexes that encounter one another on the same DNA in living cells bypass each other or collide (*Anchimiuk et al., 2021*; *Brandão et al., 2021*). Cohesin and condensin are the most characterized SMC complexes found in many eukaryotes. In bacteria, three different forms of SMC-like complexes defined as bacterial condensins have been identified, Smc-ScpAB, MukBEF, MksBEF; they are considered functionally related to condensins as they are thought to compact chromosomes and facilitate the segregation of sister chromosomes (*Lioy et al., 2018*; *Lioy et al., 2020*; *Marbouty et al., 2015*). In bacteria, Smc-ScpAB represents the most highly conserved complex, while MukBEF and MksBEF represent diverged SMC complexes (*Cobbe and Heck, 2004*; *Yoshinaga and Inagaki, 2021*).

Eukaryotic and prokaryotic SMC complexes are composed of at least five subunits: two Smc subunits, a kleisin subunit, and two additional subunits (referred to as kite and hawk subunits, depending on the type of SMC complex). Smc proteins associate with the kleisin protein to form a ring-shaped ATPase assembly. Additional subunits associate with this tripartite complex: the 'Kite' family associates with bacterial and archaeal SMC complexes and also with the eukaryotic SMC5/6 complex while the 'Hawk' family interacts with condensin and cohesin (*Yatskevich et al., 2019*). These additional subunits are thought to be required for the activity and to differentiate functions; for example, while condensin I and II share the same pair of Smc proteins, the difference in the subunit composition specifies their spatiotemporal dynamics and functional contributions to mitotic chromosome assembly (*Hirota et al., 2004*; *Kong et al., 2020*).

SMC function and dynamics on DNA requires additional auxiliary proteins (*Baxter et al., 2019*). More specifically, its loading on DNA and unloading of the DNA may depend on specific factors, at specific sites. For example, in *Bacillus subtilis*, the segregation ParB protein bound to *parS* site directly binds the Smc subunit; the ParB clamp presumably presents DNA to the SMC complex to initiate DNA loop extrusion (*Gruber and Errington, 2009*; *Sullivan et al., 2009*; *Wang et al., 2015*; *Wang et al., 2017*). Upon translocation, the site-specific recombinase XerD bound to its binding site unloads SMC complexes in the terminus region of the chromosome and this process is thought to involve specific interactions between the different components (*Karaboja et al., 2021*).

MukBEF was the first SMC complex identified. In *Escherichia coli*, MukBEF is thought to be required for chromosome segregation as *muk* mutants present many anucleate cells or mis-segregated chromosomes (*Niki et al., 1991*). How MukBEF may promote chromosome segregation and organization has remained elusive for a long time and is still not clear (*Nolivos and Sherratt, 2014*). The effect of the MukBEF complex in *E. coli* appears to be radically different from that of SMC in *B. subtilis* or other bacteria. Instead of aligning the chromosome arms from a centromere-like locus, MukBEF promotes DNA contacts in the megabase range within each replication arm (*Lioy et al., 2018*; *Lioy et al., 2020*). MukBEF promotes long-range interactions along the chromosome except in the 800-kb-long terminal domain (Ter) where MatP prevents its activity (*Lioy et al., 2018*). Under conditions of increased chromosome occupancy of MukBEF, the *E. coli* chromosome appears to be organized as a series of loops around a thin (<130 nm) MukBEF axial core (*Mäkelä and Sherratt, 2020*). Whether MukBEF is loaded at a particular locus is still an open question. The complete atomic structure of MukBEF in complex with MatP and DNA has been determined by electron cryomicroscopy (*Bürmann et al., 2021*); it also contains the MukBEF binding partner AcpP protein (*Prince et al., 2021*). It revealed that the complex binds two distinct DNA double helices reminiscent of the arms of an extruded loop, MatP-bound DNA threads through the MukBEF ring, while the second DNA is clamped by MukF, MukE, and the MukB ATPase heads. The presence of MatP in the complex together with its ability to prevent MukBEF activity prompted authors to propose that MatP might be an unloader of MukBEF (*Bürmann et al., 2021*).

Here, we have performed a number of experiments using different in vivo approaches to further characterize how MukBEF contributes to chromosome management in *E. coli*. We have used ChIP-seq and Chromosome Conformation Capture (3C-seq and Hi-C) experiments to study how MukBEF is loaded on the chromosome and promotes long-range DNA contacts. By using strains with various chromosome configurations obtained by programmed genetic rearrangements, we have explored how MukBEF activity proceeds along the chromosome and how MatP bound to *matS* sites prevents

its activity. Our results together with comparative genomics analyses allow us to address the biological significance of multiple *matS* sites in the Ter region of chromosomes in enterobacteria and of the absence of MukBEF activity.

## Results

### MukBEF activity does not initiate at a single locus

In order to characterize how MukBEF interacts with the chromosome and initiate its activity of long-range contacts, it was necessary to set up a system to reveal using Hi-C the appearance and spread of long-range contacts along the chromosome upon MukBEF synthesis. The rationale was based on previous findings showing that long-range DNA contacts within replication arms, outside Ter, result from MukBEF activity. If MukBEF loads at a specific locus as observed for Smc-ScpAB at *parS* sites, we would expect to detect the appearance and spreading of long-range contacts from this site upon MukBEF synthesis. By contrast, if MukBEF loads stochastically or at multiple loci on the chromosome, long-range contacts should occur at multiple sites.

The analysis of MukBEF activity required an efficient system to control its activity. To conditionally synthesize MukBEF, the *mukBEF* operon was cloned onto a medium-copy number plasmid under control of a p*Lac* promoter and introduced in a *mukF* mutant. In the absence of inducer, as observed for a *mukF* mutant, no growth was detected at 37°C and the amount of anucleated cells at 22°C was similar to that of the *mukF* mutant (13% vs 15%). Induced expression of *mukBEF* functionally complemented the absence of MukF, restoring growth at 37°C in Lennox Broth (LB) (*Figure 1A*) and accurately segregating the chromosome at 22°C, as evidenced by the low amount of anucleate cells (<2%) observed in the presence of the inducer (*Figure 1B*).

To determine how MukBEF activity initiates in the *E. coli* chromosome, we induced the expression of *mukBEF* and monitored the appearance of long-range DNA contacts at different times after induction (*Figure 1*). MukBEF synthesis was monitored by western blot (*Figure 1—figure supplement 1D, E*). Hi-C contact maps were established 20min, 40min, and 2 hr after induction and compared to that obtained in the absence of induction (*Figure 1C* with *Figure 1—figure supplement 1B, C*): long-range DNA contacts were hardly visible after 20 min (*Figure 1C* with *Figure 1—figure supplement 1A*), were readily observed after 40 min, and reached after 2 hr a level similar to that observed in wild-type (WT) strains (*Figure 1* with *Figure 1—figure supplement 1C*). The ratio of normalized contact maps of the induced strain at different time points to the non-induced strain allowed to visualize the presence of long-range contacts all over the chromosome except in Ter (*Figure 1D*).

The range of DNA contacts along the chromosome was quantified by measuring the width of the diagonal perpendicular to it, using an adapted version of the quantification method developed before (*Wang et al., 2017*; *Lioy et al., 2020*). The plot obtained (*Figure 1E*) allows to estimate the effect of MukBEF activity on all loci of the *E. coli* genome. This confirmed the increase in the range of contacts following induction of MukBEF predicted from the ratio of normalized contact maps, with 40 min necessary to observe a significant effect and 2 hr to reach the maximum range of contact, as observed in the WT strain (*Lioy et al., 2020*; *Figure 1* with *Figure 1—figure supplement 1*). As expected, Ter region was not affected by the induction of MukBEF because of the presence of MatP bound to *matS* sites.

Altogether, the results indicate that the increase of contacts does not originate from a specific position on the chromosome but rather appears from numerous sites, suggesting that the activity of MukBEF does not initiate at a single locus but rather from multiple loci along the chromosome.

### MukBEF activity initiates in different regions of the *E. coli* genome

The MatP/*matS* system has been shown to prevent MukBEF activity in Ter (*Lioy et al., 2018*). We took advantage of this property to unravel mechanistic aspects of MukBEF activity and address the following issues: could MukBEF interacts with a region flanked by different Ter segments and could MukBEF operate from different regions of the chromosome. To answer these questions, we used bacteriophage λ site-specific recombination as described before (*Thiel et al., 2012*) to perform large transpositions of a segment of the right ('RiTer') and left ('LiTer') replichores at different loci (*Figure 2—figure supplement 1A*), in Ter, and analyze MukBEF activity in the resulting strain (*Figure 2A*).

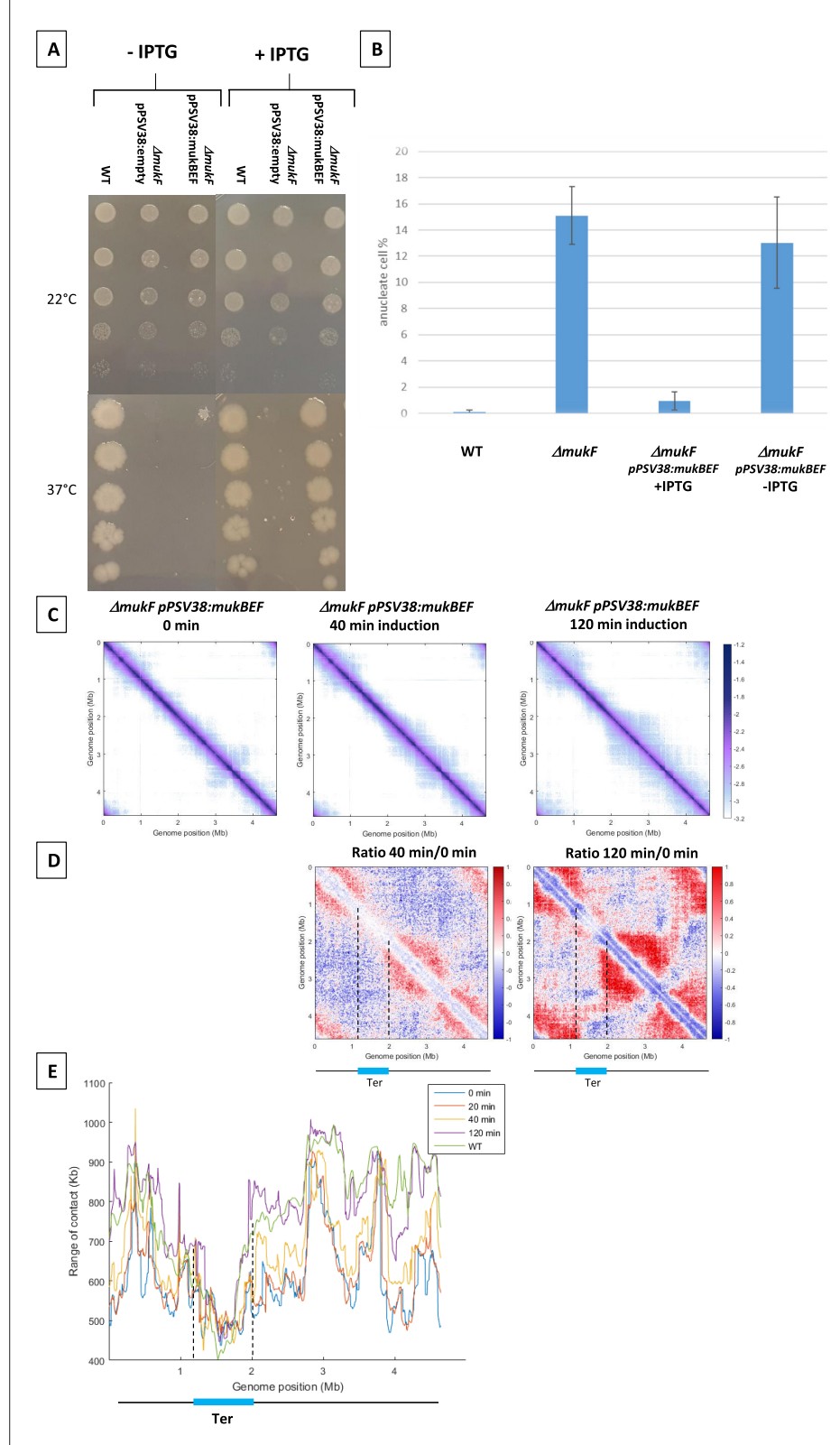

**Figure 1.** MukBEF activity detected along the chromosome. (**A**) Complementation of the Δ*mukF* mutant. Serial dilutions (10⁻¹ to 10⁻⁵) of an exponential culture of MG1655 (wild-type [WT]), MG1655 Δ*mukF* ppSV38, and MG1655 Δ*mukF* ppSV38::*mukBEF* were plated on media with (right) or without (left) inducer (IPTG). Plates were incubated at a permissive temperature of 22°C (top) or at a non-permissive temperature of 37°C (bottom). (**B**) Percentage of

*Figure 1 continued on next page*

*Figure 1 continued*

anucleate cells (blue bars) in WT and *mukF* mutant strains, complemented or not by the plasmid ppsV38::mukBEF, grown in minimal medium at 22°C with or without IPTG. Anucleate cells were identified and counted after DAPI staining. The histograms and error bars represent the means and standard deviations from at least three independent experiments. (**C**) Normalized Hi-C contact maps (5 kb bin resolution) obtained from a *ΔmukF* strain complemented with ppsV38::*mukBEF* after different induction times (0, 40, 120 min). Cells were grown in permissive conditions at 22°C in minimal medium. The x- and y-axes represent genomic coordinates in megabases (Mb). Dashed lines indicate the Ter position. (**D**) Ratio of normalized contact maps of *ΔmukF* ppSV38::*mukBEF* grown in the presence and absence of ITPG, represented in the left panel for 40 min of growth. The right panel shows the ratio of normalized contact maps for 2 hr of growth with and without ITPG. A decrease or increase in contacts in the induced condition compared with the non-induced condition is represented with a blue or red color, respectively. The black line represents a schematic chromosome, with the Ter domain highlighted in light blue. Dashed lines indicate the Ter position. (**E**) Quantification of the range of *cis* contacts of chromosomal loci along the chromosome of a WT strain grown at 22°C (green line) and of a *ΔmukF* derivative carrying ppsV38::*mukBEF* grown at 22°C under four different conditions: light blue (without ITPG), red (after 20 min of ITPG), yellow (after 40 min of ITPG), and purple (after 2 hr of ITPG). Dashed lines indicate the Ter position.

The online version of this article includes the following source data and figure supplement(s) for figure 1:

**Figure supplement 1.** Hi-C matrix of the mukBEF-complemented strain.

**Figure supplement 1—source data 1.** Original file for the Western blot analysis in *Figure 1—figure supplement 1D* anti-FLAG with MukB-Flag.

**Figure supplement 1—source data 2.** Original file for the Western blot analysis in *Figure 1—figure supplement 1D* anti-α subunit of the RNA polymerase.

**Figure supplement 1—source data 3.** pdf containing *Figure 1—figure supplement 1C* and original scans of relevant western blot analysis.

The transposition in the RiTer11 configuration results in a region of 450 kb devoid of *matS*, located between two Ter segments of 378 and 437 kb, which contain 11 *matS* and 17 *matS*, respectively (*Figure 2—figure supplement 1A*). 3C-seq experiments were performed with this strain ('RiTer11') and on the same strain deleted for *matP* or for *mukF* (*Figure 2A* with *Figure 2—figure supplement 1B*). The ratio of normalized contact maps of the RiTer11 strain to the RiTer11 *ΔmatP* allowed to visualize the reduction of long-range contacts in the two Ter segments in the presence of MatP (*Figure 2B*). On the other hand, the ratio of normalized contact maps of the RiTer11 strain to the RiTer11*ΔmukB* allowed to visualize the increase of long-range contacts between and outside the two Ter segments in the presence of MukBEF. Remarkably, long-range DNA contacts specific of MukBEF activity were observed over the chromosome except in the two Ter segments. The range of DNA contacts along the chromosome was quantified by measuring the width of the diagonal in the two strains; these plots (*Figure 2C*) allow to estimate the effect of MukBEF activity on all loci inside and outside Ter segments. These results indicated that MukBEF can operate to form long-range contacts between the two Ter segments and that MatP can prevent MukBEF activity when segments carrying *matS* sites are moved in different locations of the genome.

Similar experiments were performed with a strain carrying a transposed segment from the left replichore in the Ter MD (LiTer 15). In this configuration, a region of 1085 kb devoid of *matS* is flanked by two Ter segments of 427 and 374 kb containing 13 *matS* and 15 *matS*, respectively (*Figure 2* with *Figure 2—figure supplement 1A*). Hi-C experiments were performed on this strain ('LiTer') (*Figure 2A*). As observed with the RiTer11 strain, long-range DNA contacts specific of MukBEF activity were observed over the chromosome except in the two Ter segments. The ratio of normalized contact maps of the LiTer15 strain to the LiTer15 *ΔmukB* allowed to visualize the increase of long-range contacts outside the two Ter segments in the presence of MukBEF (*Figure 2B and C* with *Figure 2— figure supplement 1B*).

Altogether, these results demonstrate that MukBEF activity can be initiated from multiple regions of the chromosome, in the right, left, or Ori regions, and that Ter segments cannot insulate DNA segments devoid of *matS* sites. They also reveal that MukBEF does not globally translocate from the Ori region to the terminus of the chromosome as observed with Smc-ScpAB in different bacteria. Furthermore, the results indicated that Ter segments carrying 17 and 11 *matS* can prevent MukBEF activity and restrict DNA contacts.

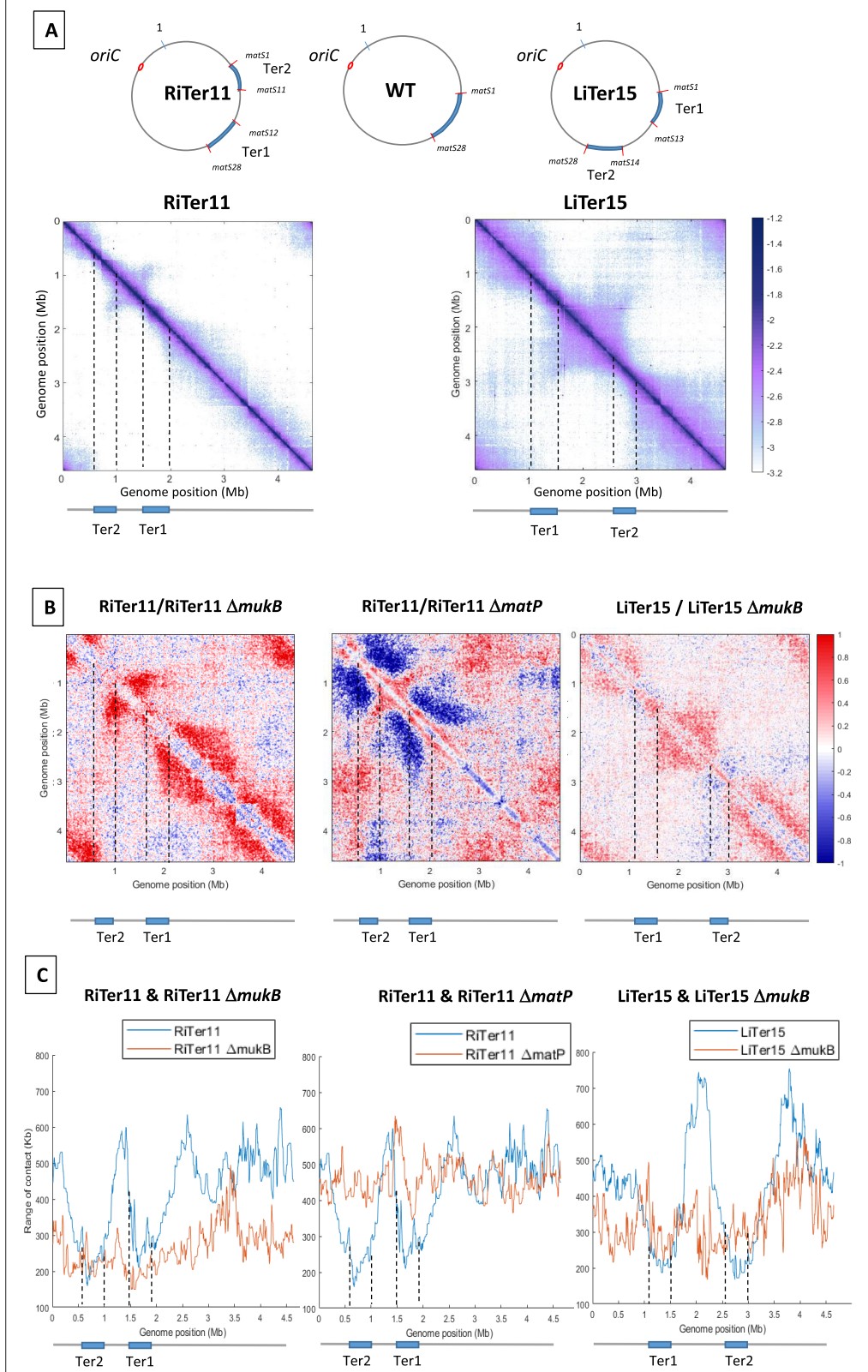

**Figure 2.** MukBEF activity initiates at different regions of the *E. coli* chromosome. (**A**) (Left) Normalized 3C-seq 5 kb bin contact map of strain RiTer11 and (right) normalized Hi-C 5 kb bin contact map of strain LiTer15. A circular representation of the chromosome for each strain has been drawn on top of each matrix, with the Ter sequence highlighted in blue and the *matS* delineating each region. Below the matrices, a linear map is provided, and

*Figure 2 continued on next page*

*Figure 2 continued*

dashed lines indicate the positions of the two Ter sequences on each matrix. (**B**) Ratio of normalized contact maps of RiTer11 strain on RiTer11 Δ*mukB* strain (left), or RiTer11 on RiTer11 Δ*matP* (middle), and LiTer15 on LiTer15 Δ*mukB* strain (right). A decrease or increase in contacts in the transposed cells compared to the transposed mutant cells is shown in blue or red color, respectively. Dashed lines indicate the Ter position. (**C**) The graph shows the quantification of the Hi-C diagonal width for loci along the chromosome of the RiTer11 strain (left panel, blue line) and the RiTer11 Δ*mukB* strain (left panel, red line), as well as LiTer15 (right panel, blue line) and LiTer15 Δ*mukB* strain (right panel, red line). Schematic representations of the two Ter segments are indicated below the graph, highlighting the fact that the range of contacts decreased on all Ter fragments. Dashed lines indicate the Ter position.

The online version of this article includes the following figure supplement(s) for figure 2:

**Figure supplement 1.** Schematic representation of transpositions and 3C-seq matrix of RiTer mukF, RiTermatp and LiTer15 mukF.

## matS determinants to prevent MukBEF activity

The method described above, splitting Ter in two parts, revealed that two parts of Ter can prevent MukBEF activity. Thus, by varying the way Ter is split in two parts, one should be able to identify *matS* determinants required to affect MukBEF activity.

Two previous studies differ slightly in the prediction for the number and consensus sequence of *matS* (*Mercier et al., 2008*; *Nolivos et al., 2016*). To clarify this, three independent ChIP-seq experiments were performed, revealing 28 *matS* and a newly derived consensus sequence (*Figure 3—figure supplement 1*). To examine the requirements for the inhibition of MukBEF by MatP, a 1 Mb region of the left replichore was transposed at different positions into the Ter domain, thus dividing Ter into two parts (referred to as Ter1 and Ter2, Ter1 comprising the *dif* site and the terminus of replication) of different sizes and carrying different number of *matS* (*Figure 3A*). Three transpositions were performed generating Ter2 domains of 253, 209, and 136 kb containing nine, seven, or four *matS*, respectively (strains LiTer9, LiTer7, LiTer4). We used Hi-C to test the ability of these Ter2 regions to inhibit MukBEF (LiTer9 in *Figure 3—figure supplement 2*, LiTer7 and LiTer4 in *Figure 3B*). Long-range DNA interactions were readily observed on both regions flanking Ter2 in both three strains, and long-range contacts are affected in the Ter2 segment, even when only four *matS* were present (*Figure 3B*). Similar Hi-C experiments were performed with the same strains deleted for *matP* (*Figure 3—figure supplement 2B*). The ratio of normalized contact maps of the LiTer4 strains to the corresponding Δ*matP* derivatives (*Figure 3C*) clearly revealed that long-range contacts were limited in Ter2 of the different strains. Remarkably, the 136 kb region carrying four *matS* sites in LiTer4 was sufficient to decrease long-range contacts promoted by MukBEF activity.

The ability of different Ter segments to inhibit the formation of long-range DNA contacts was quantified by measuring the range of contacts in the different Ter segments (*Figure 3D and E*). The median range of contact for Ter in the WT condition is 263±24 kb. Remarkably, no significant differences in the range of contacts (275±9 kb) were observed in strain Liter9 for the Ter2 segment carrying nine *matS*; it is noteworthy that this region carrying nine *matS* sites shows the same range of contacts (280±35 kb) in the WT configuration. This result shows that a transposed 253 kb region carrying nine *matS* prevent MukBEF activity as much as the same segment present in the 838-kb-long Ter MD.

To further characterize how MatP/*matS* can prevent MukBEF activity, we measured the range of contacts in Ter2 segments carrying seven or four *matS* sites. For the 209 kb segment carrying seven *matS* sites, the range of contact was increased to 330±9 kb while it increased to 352±12 kb for the 136 kb segment with four *matS*. Values obtained for these segments when present as part of the WT Ter were slightly lower (*Figure 3E*). These results indicate that segments carrying seven or four *matS* affect MukBEF activity, even though at a lower level than a 250 kb segment with nine *matS* sites.

The four *matS* sites present in the 136 kb Ter2 region of LiTer4 are not distributed regularly in that region (*Figure 3A*). We took advantage of this irregular spacing to test the capacity of a segment carrying three *matS* sites to prevent MukBEF activity and to explore whether the density of *matS* sites can modulate MukBEF activity. To address these two questions, we deleted either *matS*[26] or *matS*[28] from strain LiTer4 and probed using Hi-C the long-range of contacts. In strain LiTer4 Δ*matS*[26], three *matS* sites are distant from 78 and 58 kb whereas the *matS* sites in LiTer4 Δ*matS*[28] strain are separated by 13 and 65 kb. The ratio of normalized contact maps of the LiTer4 Δ*matS*[26] or LiTer4 Δ*matS*[28]

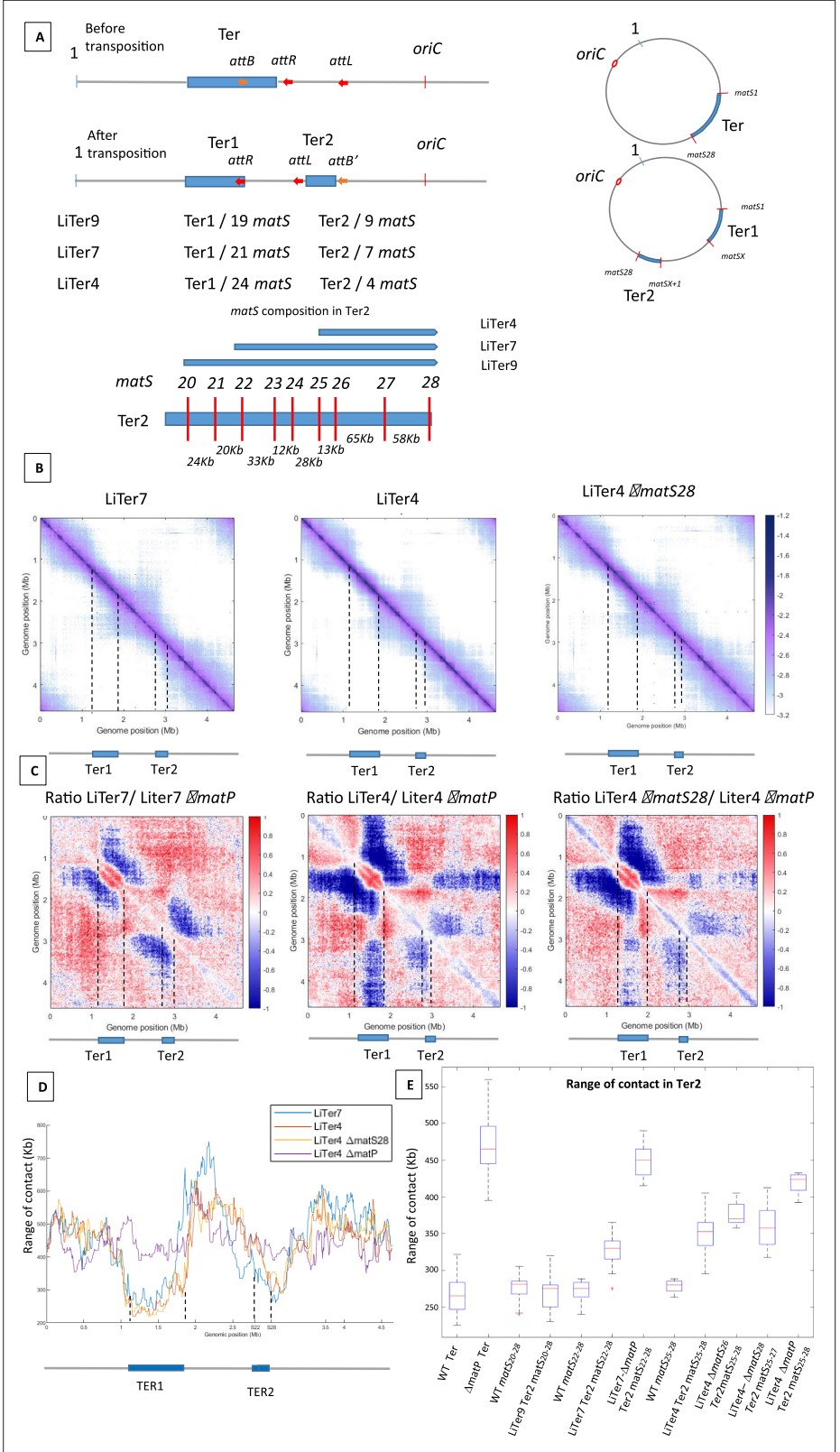

**Figure 3.** The number and distribution of *matS* sites have different effects on MukBEF inhibition. (**A**) The figure shows a schematic representation of different transpositions with the number of *matS* sites located on each Ter region. The three *att* sites are integrated into the chromosome in the same orientation, and *attL*/*attR* are fixed positions on all LiTer transpositions. *attB* is inserted at different positions into the Ter allowing, upon transposition,

*Figure 3 continued on next page*

*Figure 3 continued*

the division of the Ter domain into two subdomains, Ter1 and Ter2, containing different numbers of *matS* sites. The number of *matS* sites is indicated for the different transpositions, and the distribution of *matS* on the Ter2 segment is indicated in the schematic for the three transposed strains, LiTer4, LiTer7, and LiTer9. (**B**) Normalized Hi-C contact map with 5 kb bin resolution of the transposed strains LiTer7, LiTer4, and LiTer4 *ΔmatS28*. The position of the different Ter regions is highlighted below the matrix, and by dashed lines on the matrix. (**C**) Ratio of normalized 5 kb bin contact maps for the different transpositions compared to the *matP* mutant on the same genetic organization. The position of the different Ter regions is highlighted below the ratio, and by dashed lines on the ratio. (**D**) Quantification of the Hi-C diagonal width for loci along the chromosome for the transposed strains LiTer7, LiTer4, the derivative mutant LiTer4*ΔmatS28*, and LiTer4*ΔmatP*. The schematic map below represents the LiTer7 configuration. The position of the different Ter regions is highlighted below the graph, and by dashed lines on the graph. (**E**) This panel quantifies the range of contacts in the Ter2 region or in the corresponding sequence on the wild-type (WT) configuration. Boxplot representations are used, indicating the median (horizontal bar), the 25th and 75th percentiles (open box), and the rest of the population. 'Ter' corresponds to the range of contacts over the entire Ter region, in the WT strain (WT Ter) or in the *ΔmatP* strain (*ΔmatP* Ter). The column WT matS[X-Y] corresponds to the range of contacts between the designated *matS* sites in the WT configuration. This portion of the Ter can be compared with the same Ter segment in the transposed strain (Ter2). Additionally, the matS[20-28] segment corresponds to Ter2 in LiTer9, just as matS[22-28] corresponds to Ter2 in LiTer7, and matS[25-28] to Ter2 in LiTer4. The range of contacts of this segment was also measured in a *ΔmatP* or *ΔmatS* background.

The online version of this article includes the following figure supplement(s) for figure 3:

**Figure supplement 1.** ChIP-seq analysis of MatP-Flag.

**Figure supplement 2.** HiC matrix of transposed strain (LiTer9, LiTer7 matP, LiTer4 derivative).

**Figure supplement 3.** HiC matrix and analysis of MatP5A.

---

strains to the corresponding *ΔmatP* derivative (**Figure 3B and C** and **Figure 3—figure supplement 2C**) revealed that long-range contacts were affected by the presence of only three *matS* sites in the 136 kb segment. By measuring the range of contacts in strains LiTer4 *ΔmatS[26]* and LiTer4 *ΔmatS[28]* (**Figure 3E**), the density of three *matS* sites (*matS25-matS26* and *matS27*) in strain LiTer4 *ΔmatS[28]* seems to be as efficient (range contact of 357±11 kb) as the four *matS* sites in strain LiTer4 (range contact of 352±12 kb). By contrast, in LiTer4 *ΔmatS[26]*, the range of contact is increased to 370±6 kb indicating a reduced ability to prevent MukBEF activity. Altogether, assuming that differences in the *matS* sequences do not modify MatP's ability to bind to the chromosome and affect its capacity to inhibit MukBEF, these results suggested that the density of *matS* sites in a small chromosomal region has a greater impact than dispersion of the same number of *matS* sites over a larger segment.

## MukBEF preferentially binds in newly replicated regions

Results presented above indicated that MukBEF activity can be initiated from multiple regions of the chromosome and that, unlike Smc-ScpAB, MukBEF does not initiate its activity in the Ori region and translocate linearly to the terminus of the chromosome. To further characterize the loading and trans-location process of MukBEF, we performed ChIP-seq experiments using a Flag version of MukB on synchronized cells using a *dnaC2* thermosensitive allele that allows to control the timing of replication initiation (**Figure 4A**). The cells were grown at a permissive temperature, then shifted to 40°C for 2 hr to allow the ongoing round of replication to complete without being able to initiate a new round; the cells were then shifted back to 30°C and samples were taken for ChIP-seq analyses after 10, 20, and 40 min. At t0, there is no variation in the number of sequences along the chromosome indicating the absence of replication. In this condition, the ChIP-seq signals show a slight enrichment of signals outside Ter. After 10 min at the permissive temperature, a fraction of >500 kb centered on *oriC* was replicated as revealed by an increase of sequencing reads in this region. After 20 min, a large zone of over 1.4 Mb was replicated, and after 40 min, the chromosome was fully replicated and the replication profile was similar to that of non-synchronized cells because of multiple new replication cycles initiated. Normalized ChIP-seq experiments were performed by normalizing the quantity of immuno-precipitated fragments to the input of MukB-Flag. This experiment showed a two- to fourfold enrichment in the regions that have been replicated (**Figure 4A** with **Figure 4—figure supplement 1**). After 10 min, the signal was increased over 500 kb on each side of *oriC*. At 20 min, the signal progressed and corresponded to the regions that have been replicated. At 40 min, when the chromosome has been

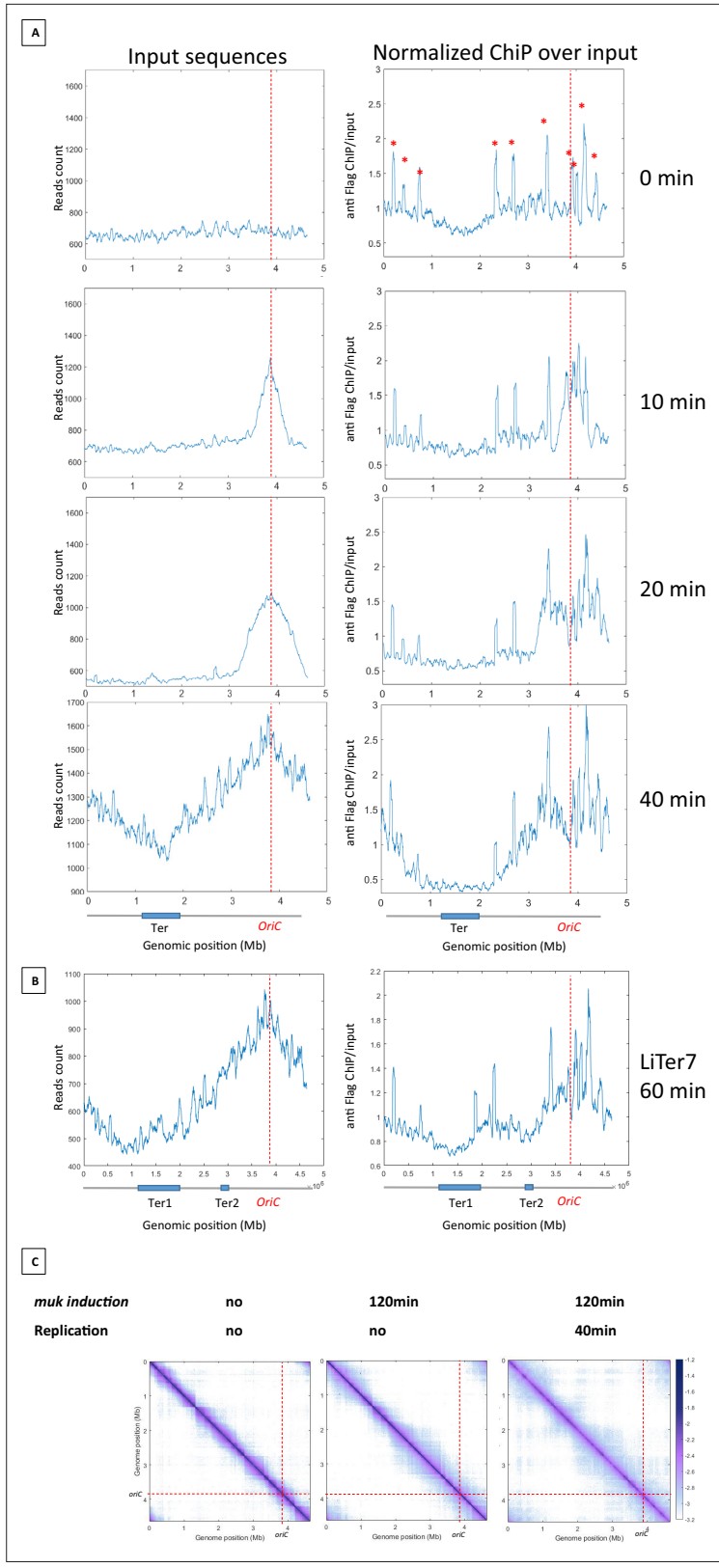

**Figure 4.** MukBEF preferentially binds in newly replicated regions and is excluded from Ter sequence. (**A**) ChIP analysis of *mukB*-Flag *dnaC2* strain. Exponentially growing cells were synchronized by incubating them at 40°C for 2 hr, followed by a shift to 30°C for 0, 10, 20, or 40 min. The replication progression was monitored by plotting the input sequence (left panel) in 50 kb bins. Normalized ChIP (ChIP over input) values for 50 kb bins are presented in

*Figure 4 continued on next page*

*Figure 4 continued*

the right panel, with the red asterisk indicating the peak observed in all *E. coli* ChIP-seq experiments. The positions of *oriC* and Ter are highlighted with a red dashed line and on the chromosome schematic below the figure. To rule out the hypothesis of non-specific antibody binding dependent on replication, we conducted a ChIP experiment without tags in synchronized cells and did not detect enrichment comparable to what is observed here (*Figure 4—figure supplement 1*). (**B**) ChIP analysis was conducted on the LiTer7 *mukB*-Flag dnaC2 strain. We observed that replication restart in this transposed strain exhibited a 20 min lag. Therefore, cells in replication stop state were shifted to 30°C for 60 min to achieve a comparable replication progression to the 40 min wild-type (WT) strain, as shown by the plotting of the input sequence (left panel). MukB enrichment generally followed the replication progression, except in the two Ter regions, as shown in the MukB normalized ChIP (right panel). (**C**) Normalized Hi-C 5 kb bin contact map of Δ*mukF* pPSV38::*mukBEF dnaC2* strain. Exponential cells were incubated at 40°C for 2 hr to prevent replication initiation (non-replicating condition) and were then shifted to 30°C for 40 min (right panel). MukBEF induction was performed by adding IPTG to the media for 2 hr. Quantification of long-range DNA contacts is shown in *Figure 4—figure supplement 3*. Red dashed lines indicate the *oriC* position.

The online version of this article includes the following figure supplement(s) for figure 4:

**Figure supplement 1.** ChIP analysis of a dnaC2 strain without the Flag tag.

**Figure supplement 2.** Domain boundaries were characterized for each condition using a DI (directionality index) analysis performed at a scale of 100 kb.

**Figure supplement 3.** Quantification of the range of cis contacts for chromosomal loci along the chromosome in the *dnaC2 ΔmukF* pPSV38::*mukBEF* strain after 2 hr at 40°C without IPTG (-rep -muk, blue line), with IPTG (-rep +muk, red line), or with IPTG followed by a transfer for 40 min at 30°C (+rep +muk, yellow line).

fully replicated, the signals obtained in the ChIP-seq samples indicated an enrichment of MukBEF all along the chromosome except in a 1.5 Mb region centered on the Ter region. As shown in *Figure 4B*, MukBEF enrichment drops to basal levels 250 kb before Ter (*Figure 4A*). This enrichment followed the progression of replication and spread from *oriC* toward Ter.

Altogether, these findings indicate that MukBEF is loaded into regions newly replicated either at the replication fork or even further behind it, except in the Ter region from which it would be excluded.

## Ter segments prevent MukBEF binding

To further explore the ability of MatP to exclude MukBEF from the Ter region, we tested the ability of MatP to exclude MukBEF from chromosomal regions containing *matS* sequences. Chip-seq experiments were performed on the Liter7 strain, which has a Ter2 segment closer to *oriC*. As in the WT background, MukBEF is preferentially associated with newly replicated sequences and shows a twofold increase following replication fork progression. However, a break in this enrichment profile is detected in the sequence corresponding to Ter2 (*Figure 4B*). These results suggest that MukBEF does not bind or persist in segments carrying *matS* sites, and that the MatP/*matS* system prevent residence of MukBEF in that region.

## MukBEF promotes long-range contacts in the absence of replication

Since MukBEF was shown to bind preferentially in newly replicated regions, we wanted to test if a preferential activity of MukBEF was detectable in those regions. To do this, we performed Hi-C on non-replicating cells lacking MukF, and induced *mukBEF* expression with or without restarting replication. In the absence of replication and of MukBEF, long-range contacts were constrained by barriers that delimit domains, called chromosome-interacting domains (CIDs) (*Figure 4C* with *Figure 4—figure supplement 2*), previously detected in WT cells in growing conditions (*Lioy et al., 2018*). Remarkably, after 2 hr of MukBEF induction in non-replicating cells, long-range contacts were detected except in the Ter region, indicating that MukBEF activity does not require newly replicated DNA to promote long-range contacts (*Figure 4C*, *Figure 4—figure supplement 3*). Finally, restarting replication for 40 min after 2 hr of MukBEF induction did not alter significantly the range and distribution of long-range DNA contacts observed in the absence of replication but in the presence of MukBEF outside Ter. As previously observed by microscopy in *E. coli,* replication does not appear to be an essential process for the activity of this SMC complex (*Badrinarayanan et al., 2012a*). Altogether, these results indicate that MukBEF promotes long-range DNA contacts independently of the replication process even though it binds preferentially in newly replicated regions.

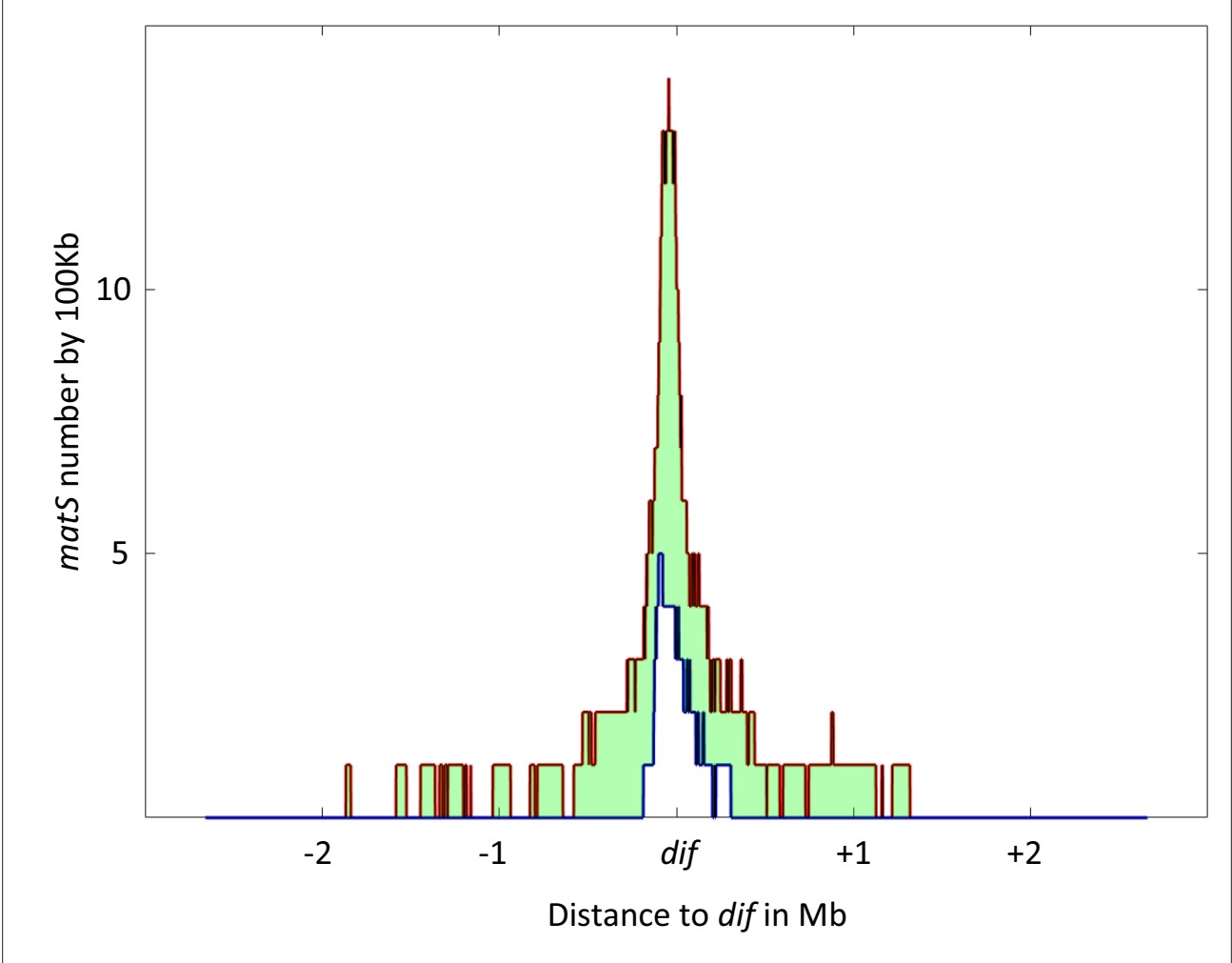

**Figure 5.** Distribution of *matS* sites culminates at *dif* across γ-proteobacteria. The density of *matS* sequences over a 100 kb unit region was measured in 16 γ-proteobacteria, and the resulting distribution is plotted in the figure. The red line represents the 75th percentile of this distribution, while the blue line represents the 25th percentile. The light green area between these two lines represents the 50% of values closest to the median. The distribution is centered on the *dif* site, and the x-axis represents the genomic distance in megabases (Mb) from *dif*, while the y-axis represents the number of *matS* sequences per 100 kilobases (kb).

The online version of this article includes the following figure supplement(s) for figure 5:

**Figure supplement 1.** Representation of *matS* density.

## Functional implications, comparative genomics of matS distribution

In *E. coli*, there are 28 *matS* sequences dispersed throughout the 1.03 Mb Ter domain. The average *matS* density is three *matS* per 100 kb, but this distribution is not uniform and the *matS* density doubles in the vicinity of the *dif* sequence (*Figure 5* with *Figure 5—figure supplement 1*). Despite the fact that only nine *matS* are sufficient to completely inhibit MukBEF activity, *matS* sites have been selected and consequently a large portion of the *E. coli* chromosome is inaccessible to MukBEF.

To determine if other species also possess a large Ter domain, we used the *matS* consensus sequence to identify the Ter domain in 16 bacteria from the enterobacterales, pasteurellales, and vibrionales groups with the higher number of sequenced genomes. The Ter domain was defined as the longest stretch of *matS* sequences between areas that are at least 100 kb devoid of *matS*. The size of the Ter domain varies from 300 kb to 1 Mb, representing 6–25% of the chromosome, and contains 6–77 *matS* (*Table 1*). In most of these species, more than 14 *matS* sequences are distributed over 540 kb, and the Ter domain always contains the *dif* sequence. To test if the *matS* distribution might differ in the vicinity of the *dif* sequence, we measured the *matS* density and centered the distribution

**Table 1.** Identification of matS sequence and Ter domain over Gammaproteobacteria.

| | Ter size* (Kb) | matS in the ter[†] | Chromsomosome size (Kb) | matS mean by 100 Kb of Ter [‡] | Size Ter/size chromosome [§] (%) |
|---|---|---|---|---|---|
| Enterobacterals | | | | | |
| *Escherichia coli* | 1037 | 31 | 4641 | 3 | 22 |
| *Salmonella enterica* serovar | 722 | 27 | 4822 | 3.7 | 15 |
| Shigella dysenteriae | 834 | 25 | 4560 | 3 | 18 |
| Klebsiella pneumoniae | 769 | 60 | 5317 | 7.8 | 14 |
| Erwinia amylovora | 435 | 19 | 3833 | 4.4 | 11 |
| Photorhabdus asymbiotica | 312 | 11 | 5064 | 3.5 | 6 |
| Pectobacterium carotovorum | 346 | 39 | 4886 | 11.3 | 7 |
| Yersinia pestis | 446 | 18 | 4658 | 4 | 10 |
| Pasteurellales | | | | | |
| Haemophilus haemolyticus | 111 | 6 | 1941 | 5.4 | 6 |
| Pasteurella canis | 332 | 11 | 2344 | 3.3 | 14 |
| Actinobacillus pleuropneumoniae | 506 | 15 | 2391 | 3 | 21 |
| Gallibacterium anatis | 350 | 6 | 2694 | 1.7 | 13 |
| Aggregatibacter actinomycetemcomitans | 506 | 15 | 2105 | 3 | 24 |
| Vibrionales | | | | | |
| Aliivibrio fischeri | 526 | 66 | 4343 | 12.5 | 12 |
| Photobacterium angustum | 409 | 77 | 4885 | 18.8 | 8 |
| *Vibrio cholerae* chr1 | 750 | 34 | 2961 | 4.5 | 25 |
| *Vibrio cholerae* chr2 | 452 | 14 | 1072 | 3.1 | 42 |
| Salinivibrio kushneri chr1 | 613 | 37 | 2840 | 6 | 22 |
| Salinivibrio kushneri Chr2 | 314 | 20 | 602 | 6.4 | 52 |

*Ter size in kilobases. Ter is defined as the longest stretch of DNA containing *matS* flanked by two regions of 100 kilobases devoid of *matS* sites.

[†]Number of matS sites identified in the Ter region using the MEME Suite and based on the matS consensus sequence (see **Figure 3—figure supplement 1**).

[‡]matS density inside the Ter region calculated as the number of matS sites divided by the size of the Ter region.

[§]Proportion of the Ter region compared to the entire chromosome, expressed as a percentage.

on the *dif* sequence. As shown in **Figure 5**, the number of *matS* per kb increases for all species and reaches its maximum near the *dif* site.

To determine if other species also possess a large Ter domain, we used the *matS* consensus sequence to identify the Ter domain in 16 bacteria from the enterobacterales, pasteurellales, and vibrionales groups with the higher number of sequenced genomes. The Ter domain was defined as the longest stretch of *matS* sequences between areas that are at least 100 kb devoid of *matS*. The size of the Ter domain varies from 300 kb to 1 Mb, representing 6–25% of the chromosome, and contains 6–77 *matS* (**Table 1**). In most of these species, more than 14 *matS* sequences are distributed over 540 kb, and the Ter domain always contains the *dif* sequence. To test if the *matS* distribution might differ in the vicinity of the *dif* sequence, we measured the *matS* density and centered the distribution on the *dif* sequence. As shown in **Figure 5**, the number of *matS* per kb increases for all species and reaches its maximum near the *dif* site.

## Discussion

SMC complexes play a fundamental role in the organization of genomes in all domains of life. Different models involving their loading, translocation, or extrusion of DNA loops as well as their unloading have been proposed, based on results obtained with different complexes and in different models. Among the SMC complexes, MukBEF has specific features: MukBEF is only found in enterobacteria and some related bacterial genera where it is involved in chromosome segregation; the mukBEF genes belong to a group of genes co-occurring with the *Dam* methylase gene including also *matP* and other genes involved in DNA metabolism (*Brézellec et al., 2006*); MukBEF exists as dimers of dimers connected by the kleisin subunit MukF (*Badrinarayanan et al., 2012b*; *Zawadzka et al., 2018*); by its activity, MukBEF does not align the two arms of the chromosome like the canonical bacterial Smc-ScpAB complex but instead promotes long-distance contacts in cis like a eukaryotic condensin (*Lioy et al., 2018*). Remarkably, MukBEF activity is not detected within the specific chromosomal Ter domain, one-fifth of the *E. coli* chromosome, due to the presence of MatP associated with this region. Although details about MukBEF and its activity have been revealed in recent years, key steps in its function remain to be characterized, including loading onto the DNA molecule, its actual loop extrusion activity, and its unloading by MatP.

Altogether, our data provide an integrated view of MukBEF activity to organize the *E. coli* chromosome. By different ChIP-seq and Hi-C approaches performed in different strains, some of which have undergone programmed genetic rearrangements, we showed that MukBEF loading does not only involve the Ori region but also different regions of the chromosome. Our results also indicated that although MukBEF binds preferentially in newly replicated regions, its activity is detected even in the absence of replication and long-range contacts appear similarly throughout the genome, except in the Ter region. These results support a model in which MukBEF molecules are bound to the chromosome, molecules are removed or displaced by replication, MukBEF molecules readily reassociate in newly replicated regions except in Ter region in which the unloading of MukBEF is enhanced by MatP bound to *matS* sites. Our results are in agreement with the previous proposal that MukBEF may organize a series of loops around a thin MukBEF axial core (*Mäkelä and Sherratt, 2020*). Altogether, our results reveal a striking contrast with the way Smc-ScpAB loads on DNA by interacting with the ParB at *parS* sites and then translocates toward the ter region (*Wang et al., 2017*). Instead, long-distance contacts promoted by MukBEF did not occur as a wave from a specific region but rather initiate at different positions, in different regions of the genome. Thus, the activity of MukBEF appears to be more similar to that of eukaryotic condensins than to that of the Smc-ScpAB complex.

The fact that MukBEF is found associated with newly replicated DNA could also suggest preferential loading at the replication fork or behind. However, in the absence of replication, MukBEF is still able to load and form long-range DNA contacts, as well as segregated chromosomes (*Badrinarayanan et al., 2012a*). Interestingly, this situation is reminiscent to that of SMC-ScpAB. Smc-ScpAB is loaded onto the *B. subtilis* chromosome at the *parS* site in association with parB/SpoJ (*Gruber and Errington, 2009*; *Sullivan et al., 2009*). Remarkably, in the absence of *parB*, Smc-ScpAB is still able to perform chromosome segregation almost perfectly (*Ireton et al., 1994*; *Wang et al., 2014*). This demonstrates that, even with a strong preferential loading mechanism, the SMC complex can load on chromosomal DNA. A similar situation may occur in *E. coli*, where MukBEF is preferentially loaded depending on the progression of replication but can still bind to and act on the chromosome in the absence of replication.

Our results showed that MukBEF was not detected in regions containing *matS* sites and that these regions were devoid of contacts extending over 600 kb. By varying the number of *matS* sites at an ectopic position, we showed that MukBEF inhibition is graded with the number of *matS*; while an effect is already visible with three *matS* scattered over a 78 kb region, the maximal effect seems to be reached with nine *matS* sites distributed over a 253 kb region. As proposed for Smc-ScpAB and its unloading XDS site (*Karaboja et al., 2021*), if the movement of DNA in the loop extrusion process involves large steps, it is conceivable that several *matS* sites are required for a complex to be trapped and discharged by MatP bound to a *matS* site. Accordingly, the density of *matS* sites seems to affect the inhibition efficiency supporting this assumption. Further experiments will be needed to analyze in detail the optimal spacing of *matS* sites to inhibit MukBEF activity.

The structure of the MukBEF-AcpP-MatP/*matS* complex obtained by cryoEM revealed the entrapment of two topologically separated DNA segments in two distinct compartments called 'ring' and

'clamp', with the *matS* site present in the ring compartment. The proposed model for MukBEF unloading stipulates that the unloading of the segment carrying *matS* site in the ring compartment is coupled to the unloading of the other segment in the clamp. One of the MatP monomers forms a contact with one of the MukE monomers while the joint binds and positions MatP between the MukB arms. The joint interface is much larger than the MukE-MatP bridge and likely provides the major binding energy for association (*Bürmann et al., 2021*). Indeed, a change to alanine of the five residues between H38 and D42 in contact with MukE did not affect MatP's ability to inhibit MukBEF activity (*Figure 3—figure supplement 3*). Refined experiments will be required to assess the outcome of mutating MatP residues involved in the interaction with the joint as mutations of those residues also affect *matS* binding (*Dupaigne et al., 2012*).

MukBEF activity is detected by the appearance of contacts over a distance of approximately 1 Mb, spanning around 4 Mb of the chromosome (excluding the Ter domain). Even if the molecular details of this activity remain to be characterized, it is tempting to speculate that MukBEF molecules can be discharged from DNA in the absence of MatP, i.e., outside Ter. Furthermore, it can be noted that MukBEF activity does not appear to be significantly disrupted in a *matP* mutant, suggesting that MukBEF can be discharged from DNA in the absence of MatP. As proposed by *Bürmann et al., 2021*, MatP would act as a structural element that ensures ideal positioning of DNA close to the exit gate in the MukBEF complex that might ensure an efficient unloading of MukBEF from the DNA. Altogether, the results would indicate that MatP is not the MukBEF unloader per se, but rather that its ability to prevent MukBEF activity has been selected to protect the Ter region from a condensin activity.

The distribution of *matS* sites over a large region of the chromosome in enterobacteria results in the inhibition of MukBEF in Ter. MatP has already been shown to confer another property to Ter, through an interaction of its C-terminus with the septum-associated protein ZapB and localizing Ter at midcell. Remarkably, these two properties are independent as inhibiting MukBEF does not require its anchoring of Ter at the septum of division (*Lioy et al., 2018*). Given that the number of *matS* sites far exceeds the number required to prevent MukBEF activity and that the density of *matS* sites increases as we approach *dif*, we may speculate that MukBEF presence is mostly banished from the *dif* region. At least two activities of DNA metabolism occur in this region: resolution of chromosome dimers by XerC-XerD recombinases and the post-replicative decatenation of circular chromosomes. We may speculate, based on in vitro observations (*Kumar et al., 2022*), that MukBEF could interfere with TopIV activity and delay potential chromosome decatenation. Another possibility is that chromosome dimers resolved at the *dif* site may become trapped in loops formed by MukBEF, thus delaying segregation. However, none of these possible scenarios are supported by data yet, and a major challenge for the future will be to determine whether and how MukBEF may interfere with one or both of these processes.

In the absence of replication and of MukBEF activity, the contact map displayed a single strong diagonal composed of very well-defined CIDs, ranging in size from 20 to 400 kb. The CIDs, clearly visible in populations of cells growing exponentially, were prominent in these conditions revealing the impact of transcription on the structuring of the genome. Upon replication, the CID organization is less apparent because of the new contacts that occur between loci belonging to different CIDs. The induction of MukBEF in the presence of replication further attenuates the patterning of CIDs. Altogether, these results highlight the respective contribution of these three processes, transcription, replication, and condensin activity, on the organization of bacterial genomes.

## Materials and methods
### Bacterial strains and plasmids
All *E. coli* strains used in this study are derived from *E. coli* MG1655. Deletion mutants were constructed as described in *Datsenko and Wanner, 2000*. Mutations were combined via P1 transduction. A plasmid capable of synthesizing MukBEF was constructed by cloning the entire *mukBEF* operon into the pPSV38 plasmid using *Eco*RI/*Xba*I. MukBEF synthesis was monitored by western blot after the addition of IPTG at 0, 20, 40, and 120 min (*Figure 1—figure supplement 1D*).

## Media and growth conditions

*E. coli* cells were cultured at 22°C and 30°C in either LB or liquid minimal medium A (MM) supplemented with 0.12% casamino acids and 0.4% glucose. When necessary, antibiotics were added to the growth media at the following concentrations: ampicillin at 100 µg/ml, kanamycin at 50 µg/ml, chloramphenicol at 15 µg/ml, spectinomycin at 50 µg/ml, apramycin at 50 µg/ml, and zeocin at 25 µg/ml.

## 3C-seq protocol

3C-seq libraries were generated as described (*Lioy et al., 2018*). Briefly 100 ml of culture were crosslinked with formaldehyde (7% final concentration) for 30 min at room temperature followed by 30 min at 4°C. Formaldehyde was quenched with a final concentration of 0.25 M glycine for 20 min at 4°C. Fixed cells were collected by centrifugation and stored at −80°C until use. Frozen pellets consisting of approximately $1-2 \times 10^9$ cells were thawed and suspended in 600 µl of TE buffer (10 mM Tris-HCl, 0.5 mM EDTA, pH 8) with 4 µl of lysozyme (35 U/µl). The mixture was incubated at room temperature for 20 min. Subsequently, SDS was added to a final concentration of 0.5% and the cells were incubated for an additional 10 min at room temperature. Lysed cells were then diluted 10 times in several tubes containing 450 µl of digestion mix (1× NEB 1 buffer, 1% Triton X-100). 100 units of *Hpa*II were added and the tubes were incubated for 2 hr at 37°C. To stop the digestion reaction, the mixture was centrifuged for 20 min at 20,000 × *g*, and the resulting pellets were resuspended in 500 µl of sterile water. The resulting digested DNA (4 ml in total) was divided into four aliquots and diluted in 8 ml of ligation buffer (1× ligation buffer NEB without ATP, 1 mM ATP, 0.1 mg/ml BSA, 125 units of T4 DNA ligase, 5 U/µl). Ligation was performed at 16°C for 4 hr, followed by overnight incubation at 65°C with 100 µl of proteinase K (20 mg/ml) and 100 µl EDTA 500 mM. DNA was then precipitated with an equal volume of 3 M Na-acetate (pH 5.2) and two volumes of iso-propanol. After 1 hr at −80°C, the DNA was pelleted and suspended in 500 µl of 1× TE buffer. The tubes were incubated directly with 50 µl of proteinase K (20 mg/ml) for an overnight period at 65°C. Subsequently, all tubes were transferred to 2 ml centrifuge tubes and extracted with 400 µl of phenol-chloroform pH 8.0. The DNA was then precipitated, washed with 1 ml of 70% cold ethanol, and resuspend in 30 µl of 1× TE buffer in the presence of RNase A (1 µg/ml). The tubes containing ligated DNA (3C libraries), digested DNA, and non-digested DNA were pooled into three separate tubes. The efficiency of the 3C preparation was evaluated by running a 1% agarose gel.

## Hi-C protocol

Hi-C libraries were generated as described (*Thierry and Cockram, 2022*).

$10^8$ cells growing in the exponential growth phase of *E. coli* were chemically crosslinked by the addition of formaldehyde directly to the cultures (3% final concentration) for 30 min at room temperature with gentle agitation. Crosslinking was quenched by the addition of glycine (0.5 M final concentration) for 20 min at room temperature. Cells were washed in 50 ml PBS 1× and centrifuged. The pellet was resuspended in 1 ml 1× PBS and transferred to a 1.5 ml Eppendorf tube before a final centrifugation step (4000 × *g*, 5 min, room temperature), the supernatant was then removed and the pellet stored at −80°C. The pellet was then resuspended in 1 ml of 1× TE + complete protease inhibitor cocktail (EDTA-free, Sigma-Aldrich) and transferred to a 2 ml Eppendorf tube with 4 µl of ready-to-lyse lysozyme for 20 min at room temperature. SDS was added to a final concentration of 0.5% and the cells were incubated for an additional 10 min. In a 10 ml Falcon tube, DNA was then prepared for digestion by the addition of 3 ml $H_2O$, 500 µl 10× Digestion buffer (200 mM Tris-HCl pH 7.5, 100 mM $MgCl_2$, 10 mM DTT, 1 mg/ml BSA) and 500 µl 10% Triton X-100 (Thermo Fisher). After thoroughly mixing the reaction, 400 µl were removed and transferred to a 1.5 ml Eppendorf tube as a non-digested (ND) control. The restriction enzyme *Hpa*II (New England Biolabs, 1000 U) was then added to the remaining sample and the tube incubated with gentle agitation for 3 hr at 37°C. The solution was centrifuged at 16,000 × *g*, 20 min, room temperature. After removing the supernatant, the pellet was resuspended in 400 µl $H_2O$, completed subsequently by adding 50 µl 10× Ligation Buffer (500 mM Tris-HCl pH 7.5, 100 mM $MgCl_2$, 100 mM DTT), 4.5 µl 10 mM dAGTTp, 37.5 µl Biotin-14-dCTP (Thermo Fisher), 50 units of DNA Polymerase I - Large Klenow Fragment (New England Biolabs). After briefly mixing, the reaction was incubated with agitation for 45 min at 37°C. The ligation was set up by adding the following: 120 µl 10× Ligation Buffer, 12 µl 10 mg/ml BSA, 12 µl 100 mM ATP, 540 µl $H_2O$, 480 U T4 DNA ligase (Thermo Fisher). The reaction was mixed gently and then incubated with gentle agitation

for 3 hr at room temperature. Following ligation, proteins were denatured by the addition of 20 μl 500 mM EDTA, 20 μl 10% SDS, and 100 μl 20 mg/ml proteinase K (EuroBio). The following day, DNA was purified using the standard procedure described for the 3C-seq.

## Chip-seq protocol

$10^9$ cells in the exponential growth phase of *E. coli* were chemically crosslinked by adding formaldehyde directly to the cultures (final concentration of 1%) for 30 min at room temperature with gentle agitation. Crosslinking was quenched by adding glycine (final concentration of 0.25 M) for 15 min at room temperature. The cells were washed twice with 1× TBS (50 mM Tris-HCl pH 7.6, 0.15 M NaCl) and the pellet was resuspended in 1 ml of 1× TBS before a final centrifugation step (4000 × *g*, 5 min, room temperature). The supernatant was then removed and the pellet was stored at –80°C.

The pellet was resuspended in 500 μl of lysis buffer 1 (20% sucrose, 10 mM Tris pH 8, 50 mM NaCl, 10 mM EDTA) and 4 μl of ready-to-lyse lysozyme was added, followed by incubation at 37°C for 30 min. Then, 500 μl of lysis buffer 2 (50 mM Tris-HCl pH 4.7, 150 mM NaCl, 1 mM EDTA, 1% Triton X-100) and a tablet of antiprotease cocktail (Roche) were added. The solution was transferred to a 1 ml Covaris tube and sonicated for 10 min (peak incident power 140 W/duty cycle 5%/cycle per burst 200). Cell debris were eliminated by centrifugation at 13,000 rpm for 30 min at 4°C. The supernatant was transferred to a fresh Eppendorf tube and 50 μl was used as input. The rest of the cell extract was mixed with 40 μl of anti-Flag M2 resin previously washed in TBS and resuspended in lysis buffer 2. The solution was incubated overnight at 4°C with gentle mixing and then centrifuged for 30 s at 5000 × *g*. The pellet was washed twice with TBS containing 0.05% Tween 20 and three times with TBS.

To elute the immunoprecipitation, 100 μl of 1× TBS containing 15 μg of 3× Flag peptide was mixed with the resin and incubated for 30 min at 4°C. Then, the solution was centrifuged for 30 s at 5000 × *g* and the supernatant was transferred to a new tube. A second elution step was performed on the resin before decrosslinking. IP and input was purified using MinElute QIAGEN columns and then sequenced.

## Processing of libraries for Illumina sequencing

The samples were sonicated using a Covaris S220 instrument to obtain fragments ranging in size from 300 to 500 base pairs. These fragments were then purified using AMPure XP beads and resuspended in 10 mM Tris-HCl. For Hi-C libraries, a biotinylated pull-down step was performed by adding 30 μl of streptavidin C1 MyOne Dynabeads from Invitrogen to 300 μl of binding buffer (10 mM Tris-HCl, pH 7.5, 1 mM EDTA, 2 M NaCl) and mixing with the Hi-C libraries for 15 min. DNA ends were then prepared for adaptor ligation following standard protocols as described in *Thierry and Cockram, 2022*.

The Illumina sequencing process was performed in accordance with the manufacturer's recommendations, with 15 cycles of amplification. The size of the DNA fragments in the libraries was assessed using TAE 1% agarose gel and tape station, followed by paired-end sequencing on an Illumina sequencer.

## Chip-seq analysis

Between 10 and 20 million reads were recovered for each sample. We used Bowtie2 software to perform mapping in local mode, and the mpileup software from Samtools to calculate coverage for each position in the genome. Normalized ChIP was then calculated by normalizing the number of reads at each position by the total number of reads, and dividing this number by the normalized input. A sliding window of 50 kb was applied to smooth the variations. Peaks for MatP ChIP-seq (*Figure 3—figure supplement 1B*) were identified by extracting positions where the number of reads was 10 times higher than the background for at least 30 consecutive base pairs. The center of these distributions was considered the center of the peak. The sequences were then extracted and used on the MEME suite to identify a common motif.

## Generation of contact maps

Contact maps were constructed as described previously (*Lioy et al., 2018*). Briefly, each read was assigned to a restriction fragment. Non-informative events such as self-circularized restriction fragments, or uncut co-linear restriction fragments were discarded, as in *Cournac et al., 2012*. The genome

was then divided into 5 kb units, and the corresponding contact map was generated and normalized through the sequential component normalization procedure of *SCN* (based on the sequential component normalization; https://github.com/koszullab/E_coli_analysis; *Koszul, 2020*; *Lioy et al., 2018*). Contact maps were visualized as logarithmic matrices to aid in visualization.

### Ratio of contact maps

The comparison of contact maps was facilitated by displaying their ratio. The ratio was calculated for each point on the map by dividing the number of contacts in one condition by the number of contacts in the other condition. The Log2 of the ratio was then plotted using a Gaussian filter. The color code represented a decrease or increase in contacts in one condition relative to the other (a blue or red signal, respectively); a white signal indicated no change.

### Quantification of the range of *cis* contacts along the chromosome

We used a three-step process adapted from *Lioy et al., 2020*, and *Wang et al., 2017*, to determine the width of the diagonal in contact maps. To improve the resolution of the Ter limit, we measured the width perpendicular to the main diagonal.

First, we calculated the median of the contact map and estimated the standard deviation ($\sigma$) using a robust statistic, where $\sigma=1.4826 * $ mad (mad = median absolute deviation). Next, we used a point connecting algorithm to differentiate significant interactions from background noise. The size of each connected element identified by the 'bwlabel' function of MATLAB was determined. We considered a connected element with a size $\geq 30$ points to be significant and used the 'imclose()' function of MATLAB to fill in the empty points within the connected elements using a diamond shape with a size of 5.

Subsequently, we calculated the width of the primary diagonal for each 5 kb bin of the genome. The range of cis contact was estimated from the width of the primary diagonal by multiplying the number of measured bins by the bin size (5 kb) and dividing by two (since the range is symmetric on both sides of the chromosomal locus being considered). Finally, a boxplot was used to visualize the entire range of *cis* contacts for all the analyzed chromosome regions.

### Directional index analysis

Directional index was calculated as described in *Lioy et al., 2018*. The directional index is a statistical metric that quantifies the level of upstream or downstream contact bias for a given genomic region (*Dixon et al., 2012*). This metric is based on a t-test comparison of contact vectors to the left and right of each bin, up to a certain scale. The boundaries between topological domains often generate fluctuating signals that result in a transition in the directional preference. Specifically, for each 5 kb bin, we extracted the contact vector from the correlation matrix between that bin and neighboring bins at regular 5 kb intervals, up to 100 kb, in both left and right directions. At each step, the paired t-test was used to determine whether the strength of interactions was significantly stronger in one direction relative to the other. A threshold of 0.05 was used to assess statistical significance. The directional preferences for each bin along the chromosome were visualized as a bar plot, with positive and negative t-values shown as red and green bars, respectively. To improve the clarity of presentation, bars for bins with t-values below –2 or above 2 (corresponding to a p-value of 0.05) were truncated. Between two identified domains in the contact matrices, the directional preference of bins changed, which was indicated by alternating red and green colors.

## Acknowledgements

We thank members of the FB laboratory for help and fruitful discussions. We also thank the I2BC NGS facility for high-throughput sequencing. This project was supported by grants from 'Agence Nationale de la Recherche' (ANR-CE12-0013-01) and the 'Association pour la Recherche contre le Cancer' (Projet Fondation ARC 2018). MS was supported by a doctoral fellowship from CNRS and from the 'Association pour la Recherche contre le Cancer'.

## Additional information

### Funding

| Funder | Grant reference number | Author |
|---|---|---|
| Agence Nationale de la Recherche | ANR-CE12-0013-01 | Frederic Boccard |
| Fondation ARC pour la Recherche sur le Cancer | PJA 20181208006 | Mohammed Seba |

The funders had no role in study design, data collection and interpretation, or the decision to submit the work for publication.

### Author contributions

Mohammed Seba, Conceptualization, Investigation, Methodology; Frederic Boccard, Conceptualization, Funding acquisition, Writing - original draft, Project administration; Stéphane Duigou, Conceptualization, Data curation, Formal analysis, Supervision, Validation, Investigation, Visualization, Methodology, Writing - original draft, Project administration, Writing - review and editing

### Author ORCIDs

Frederic Boccard (ID) http://orcid.org/0000-0002-1866-2489
Stéphane Duigou (ID) http://orcid.org/0000-0001-5103-4663

Reviewer #1 (Public Review): https://doi.org/10.7554/eLife.91185.3.sa1
Reviewer #2 (Public Review): https://doi.org/10.7554/eLife.91185.3.sa2
Reviewer #3 (Public Review): https://doi.org/10.7554/eLife.91185.3.sa3
Author Response https://doi.org/10.7554/eLife.91185.3.sa4

## Additional files

### Supplementary files
• Supplementary file 1. Table of strains.

• MDAR checklist

### Data availability

The DNA sequencing raw data was deposited on the NCBI website under the bioproject numbers: PRJNA1019269 (HiC induction of MukBEF), PRJNA1021387 (HiC on transposed strains), PRJNA1024671 (HiC on non-replicating strains), and PRJNA1024669 (ChIP-seq of MukB on synchronized strains).

The following datasets were generated:

| Author(s) | Year | Dataset title | Dataset URL | Database and Identifier |
|---|---|---|---|---|
| Seba M, Boccard F, Duigou S | 2023 | HiC Ecoli MukBEF induction | https://www.ncbi.nlm.nih.gov/bioproject/PRJNA1019269/ | NCBI BioProject, PRJNA1019269 |
| Seba M, Boccard F, Duigou S | 2023 | HiC transposed Ter strain | https://www.ncbi.nlm.nih.gov/bioproject/?term=PRJNA1021387 | NCBI BioProject, PRJNA1021387 |
| Seba M, Boccard F, Duigou S | 2023 | HiC +/- repliction | https://www.ncbi.nlm.nih.gov/bioproject/?term=PRJNA1024671 | NCBI BioProject, PRJNA1024671 |
| Seba M, Boccard F, Duigou S | 2023 | MukB ChIP-seq on synchronized cells | https://www.ncbi.nlm.nih.gov/bioproject/?term=PRJNA1024669 | NCBI BioProject, PRJNA1024669 |

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
