## [Editor Report · eLife assessment]

This **important** work combines DNA contact analysis and controlled genome rearrangements to investigate the processes that organize the *E. coli* chromosome, with a particular focus on how the SMC-related complex MukBEF is regulated. The evidence supporting the conclusions is **compelling**, with time-resolved experiments and analysis of mutant strains. The work will be of broad interest to chromosome biologists and bacterial cell biologists.

---

## [Referee Report · Reviewer #1 (Public Review)]

In this manuscript, Seba et al., investigate the mechanism of chromosome organization by the MukBEF complex in *E. coli*. They use a combination of Hi-C and ChIP analysis to understand the steps of MukBEF regulation: its unloading from DNA (how MukBEF activity is prevented in the terminus regions of the chromosome by MatP), and its loading onto DNA (how DNA replication influences MukBEF association with the chromosome). Seba et al., induce chromosomal rearrangements to flip the sections of the ter region, thus perturbing matS site numbers and position. They find that MukBEF activity is prevented around matS sites and that higher matS density has greater effect on MukBEF. Separately, using replication mutants and inducible MukBEF expression, they find that MukBEF can associate with the chromosome even in the absence of replication (as seen by the emergence of long-range contacts). However, ChIP data suggests that MukBEF binding to DNA is enriched on newly replicated DNA.

Altogether, this work provides a valuable and comprehensive view of MukBEF-mediated chromosome organization, with insights on the mechanism of the exclusion of MukBEF from the terminus region of the chromosome. The use of the programmed genetic rearrangements is powerful and allows the authors to provide clear and convincing evidence for MukBEF exclusion from ter by matS sites. It is particularly striking to see that MukBEF can promote long-range contacts even in chromosomal regions between two matS, but the complex is excluded from the matS 'zones'. Experiments using cells blocked for replication show that MukBEF can influence chromosome organization in the absence of replication as well. While previous studies have reported some evidences in support of both of the above conclusions, the experiments described here offer a clear and direct demonstration of the same.

---

## [Referee Report · Reviewer #2 (Public Review)]

Summary:

Chromosome organization in *E. coli* and related species ('transversal') deviates starkly from the pattern more commonly found in bacteria ('longitudinal'). The underlying mechanisms and the physiological roles, however, are not well understood. This manuscript by Seba et al. investigates the activity and regulation of MukBEF in chromosome folding in *E. coli*. Using a construct for inducible expression of MukBEF, the authors first demonstrate that the initiation of long-range chromosome contacts (likely by loop extrusion) is not restricted to few positions on the chromosome and rather widely distributed but excluding the replication terminus region. Using ChIP-Seq, the authors show that the distribution of MukBEF over the chromosome is consistent with widely distributed loading and moreover indicate a connection of chromosome folding and DNA replication with newly replicated DNA shower an increased tendency for MukBEF binding. To dissect this further, they then redistribute matS sites on the chromosome by a clever strategy based on large-scale transpositions. The results reveal that matS-free DNA segments undergo MukBEF dependent folding regardless of their position relative to the origin of replication, being consistent with a broad distributed loading of MukBEF. By fine-mapping with smaller transposition events, they show that few matS sites are sufficient to impede MukBEF activity. Surprisingly, however, *E. coli* and most related genomes harbor many matS sites, which are particularly highly concentrated near the chromosome dimer resolution dif site (Fig. 5).

This is a well-executed and well-presented study. The findings show that the MatP/matS system acts locally and independent of DNA replication to restrict MukBEF in the replication terminus region. Few of the many matS sites are sufficient for MukBEF restriction. The main conclusions of the work are clear and well supported by the data.

---

## [Referee Report · Reviewer #3 (Public Review)]

Seba et al. investigate whether chromosomal recruitment of the *E. coli* SMC complex MukBEF is initiated at a single site, how MukBEF activity is excluded from the replication terminus region, and whether its recruitment and activity depend on DNA replication. Upon induction of MukBEF, the authors find that chromosomal long-range contacts increase globally rather than from a single site. Using large-scale chromosome rearrangements, they show that matS sites can insulate separate areas of high MukBEF activity from each other. This suggests that MukBEF loads at multiple sites in the genome. Finally, the authors propose that MukBEF associates preferentially with newly replicated DNA, based on ChIP-seq experiments after DNA replication arrest.

The conclusions of the paper are well supported by the data. The ratiometric contact analyses and range-of-contact analyses are compelling and nicely show the interplay between MukBEF and its proposed unloader MatP/matS. I particularly enjoyed the chromosome re-arrangement experiments, which lend strong support to the idea that MukBEF activity is independent of a centralized loading site.

The enrichment of MukBEF in newly replicated regions is convincing, despite somewhat small effect sizes. The suggestion that matS density controls MukBEF activity is appealing, but will need additional support from more systematic studies. It is based on a comparison of only two strains (looking at different combinations of three matS sites), and the effect size is small. As it is, differences in matS sequence composition and genomic context cannot be factored out.

Overall, the work is an important advance in our understanding of bacterial chromosome organization. It will be of broad interest to chromosome biologists and bacterial cell biologists.

---

## [Author Response]

The following is the authors’ response to the original reviews.

**Reviewer #1 (Recommendations For The Authors):**
1. Experiments regarding the inducible expression of MukBEF: The authors should provide western blots or rt-qPCR for MukBEF expression at 40 min and 2H.

We provide now a western blot of MukB in non-induced and induced conditions as Figure 1-figure supplement 1D.

1. Experiments with RiTer and LiTer constructs:

We agree that the matP mutant may help the reader to compare the effect of the translocation in different backgrounds and have added it to the figure. This strengthens the conclusion that longrange interactions in ter do increase in the absence of matP in a rearranged chromosome, as observed in the WT configuration (Lioy et al., 2018).

b. Additionally, in Fig. 2C, it appears that there is some decrease in long-range interactions in the absence of mukB in ter1 (Riter). Is this a significant change?

The change observed is not significant. The results shown in Fig. 2C have been obtained using a 3C approach, which generated slightly more variability than Hi-C. Furthermore, we measured the range of contacts for the segment corresponding to Ter1 in RiTer (matS12-matS28), in different genetic contexts and different configurations. The results show that this level of variation is not significant (see graph below reporting two independent experiments).

**Author response image 1. sa4fig1:** Range of interactions measured on the interval matS12-matS18 in different genetic contexts and different configurations (MG1655 WT(1 and 2), ∆mukB, RiTer, RiTer ∆mukB).

1. Experiments with various matS organizations: These experiments are interesting and an important part of the paper. However, it is rather hard to visualize the chromosome conformations in the strains after transposition. To aid the reader (particularly with panel E), authors can provide schematics of the chromosome conformations and anticipated/ observed chromosomal interactions. Circular interaction plots would be useful here.

We thank the reviewer for this interesting remark; we have tried in the past to represent these interactions using a circular representation (see for example the web site of Ivan Junier; https://treetimc.github.io/circhic/index.html). However, this representation is not trivial to apprehend for nonspecialists, especially in strains with a rearranged chromosome configuration. Nonetheless, we have added graphical circular representations of the chromosome configurations to help the reader.

1. ChIP experiments:

The basal value of the ChIP on the non-replicated sequences (between 0-3.5 Mb for 10 minutes and 0-3 Mb for 20 minutes) is 0.8 and 0.7, respectively, whereas the mean value of the replicated sequence is 1.6 and 1.45. So the enrichment observed for these two points is about 2-fold, not 1.1 and it is 4 fold for t40min.These values were obtained by dividing the number of normalized reads in the ChIP (the number of reads at each position divided by the total number of reads) by the normalized reads of the input. Therefore, the increase in copy number is considered in the calculation. Furthermore, we added a supplementary figure (Figure Sup9) in which we performed a ChIP without tags on synchronized cells, and in this case, we did not observe any enrichment triggered by replication.

b. Authors make a conclusion that MukB loads behind the replication fork. However, the time resolution of the presented experiments is not sufficient to be certain of this. Authors would need to perform more time-resolved experiments for the same.

Reviewer 1 is correct; we attempted to discriminate whether the observed enrichment is (i) associated with the replication fork since we observed a decrease in the center of the enrichment at oriC as the maximum enrichment moves away with the replication fork after 20 and 40 minutes, or (ii) associated with the newly replicated sequence. To investigate this, we attempted to induce a single round of replication by shifting the cells back to 40°C after 10 minutes at 30°C. Unfortunately, replication initiation is not immediately halted by shifting the cells to 40°C, and we were unable to induce a single round of replication. To clarify our conclusions, we modified our manuscript to

“Altogether, these findings indicate that MukBEF is loaded into regions newly replicated either at the replication fork or even further behind it, except in the Ter region from which it would be excluded.”

c. Authors conclude that in the LiTer7 strain, MukB signal is absent from Ter2. However, when compared with the ChIP profiles by eye across panels in A and B, this does not seem to be significant. In the same results sections, authors state that there is a 3-fold increase in MukB signal in other regions. The corresponding graph does not show the same.

Rather than relying solely on the enrichment levels, which can be challenging to compare across different strains due to slight variations in replication levels, we believe there is a clear disruption in this profile that corresponds to the Ter2 sequence. Furthermore, this discontinuity in enrichment relative to the replication profile is also observable in the WT configuration. At T40min, MukB ChIPseq signals halt at the Ter boundary, even though Ter is actively undergoing replication, as evidenced by observations in the input data.

Regarding the fold increase of MukB, Reviewer 1 is correct; we overestimated this enrichment in the text and have now corrected it.

d. Authors should provide western blot of MukB-Flag.

We have added Supplementary Figure 1 D, which contains a Western blot of MukB-Flag.

1. The bioinformatic analysis of matS site distribution is interesting, but this is not followed upon. The figure (Fig 5) is better suited in the supplement and used only as a discussion point.

We acknowledge the reviewer's point, but we used this section to attempt to extend our findings to other bacteria and emphasize the observation that even though a few matS sites are necessary to inhibit MukBEF, the Ter domains are large and centered on dif even in other bacteria.

1. The discussion section is lacking many references and key papers have not been cited (paragraph 1 of discussion for example has no references).The possibility that SMC-ScpAB and MukBEF can act independent of replication has been suggested previously, but are not cited or discussed. Similarly, there is some evidence for SMC-ScpAB association with newly replicated DNA (PMID 21923769).

We have added references to the suggested paragraph and highlighted the fact that MukBEF's activity independent of replication was already known. However, we believe that the situation is less clear for SMC-ScpAB in *B. subtilis* or C. crescentus. In a similar manner, we found no clear evidence that SMCScpAB is associated with newly replicated DNA in the referenced studies.

To clarify and enrich the discussion section, we have added a paragraph that provides perspective on the loading mechanisms of SMC-ScpAB and MukBEF.

1. There are minor typographical errors that should be corrected. Some are highlighted here:a. Abstract: L5: "preferentially 'on' instead of 'in'"b. Introduction: Para 1 L8: "features that determine"c. Introduction: Para 2 L1: please check the phrasing of this lined. Results section 2: L1: Ter "MD" needs to be explainede. Page 8: Para 2: L6: "shows that 'a'"g. Page 13: Para 2: "MukBEF activity...". This sentence needs to be fixed.i. Figure 4: "input" instead of "imput"

We thank Reviewer 1 for pointing out all these grammatical or spelling mistakes. We have corrected them all.

f. Page 12: Para 2: "Xer" instead of "XDS"? *We added a reference to clarify the term.h. Methods: ChIP analysis: Authors state "MatP peaks", however, reported data is for MukB

This description pertains to the matP peak detection shown in Supplementary Figure 3. We have incorporated this clarification into the text.

j. Supplementary figure legends need to be provided (currently main figure legends appear to be pasted twice)

Supplementary figure legends are provided at the end of the manuscript, and we have edited the manuscript to remove one copy of the figure legends.

k. Authors should ensure sequencing data are deposited in an appropriate online repository and an accession number is provided.We waited for the appropriate timing in the editing process to upload our data, which we have now done. Additionally, we have added a data availability section to the manuscript, including sequence references on the NCBI.
**Reviewer #2 (Recommendations For The Authors):**
The authors largely avoid speculation on what might be the physiological relevance of the exclusion of MukBEF (and Smc-ScpAB) from the replication termination region (and the coordination with DNA replication). At this stage it would be helpful to present possible scenarios even if not yet supported by data. The authors should for example consider the following scenario: loop extrusion of a dif site in a chromosome dimer followed by dimer resolution by dif recombination leads to two chromosomes that are linked together by MukBEF (equivalent to cohesin holding sister chromatids together in eukaryotes but without a separase). This configuration (while rare) will hamper chromosome segregation. Is MatP particularly important under conditions of elevated levels of chromosome dimers? Could this even be experimentally tested? Other scenarios might also be entertained.

Even though we prefer to avoid speculations, we agree that we may attempt to propose some hypotheses to the reader. To do so, we have added a few sentences at the end of our discussion. “We may speculate, based on in vitro observations (Kumar et al., 2022), that MukBEF could interfere with TopIV activity and delay potential chromosome decatenation. Another possibility is that chromosome dimers resolved at the dif site may become trapped in loops formed by MukBEF, thus delaying segregation. But none of these possible scenarios are supported by data yet, and a major challenge for the future is to determine whether and how MukBEF may interfere with one or both of these processes.”

The manuscript text is well written. However, the labeling of strains in figures and text is sometimes inconsistent which can be confusing (LiTer Liter liter; e.g Riter Fig 2C). For consistency, always denote the number of matS sites in LiTer strains and also in the RiTer strain. The scheme denoting LiTer and RiTer strains should indicate the orientation of DNA segments so it is clear that the engineering does not involve inversion (correct?). Similarly: Use uniform labelling for time points: see T40mn vs 40mn vs T2H vs 2H

We have reviewed the manuscript to standardize our labeling. Additionally, we have included a schema in Figure 2, indicating the matS numbers at the Ter border to emphasize that the transposition events do not involve inversion.

matS sites do not have identical sequences and bind different levels of MatP (suppl fig 3). Does this possibly affect the interpretation of some of the findings (when altering few or only a single matS site). Maybe a comment on this possibility can be added.

We agree with the referee; we do not want to conclude too strongly about the impact of matS density, so we have added this sentence at the end of the section titled 'matS Determinants to Prevent MukBEF Activity':

“Altogether, assuming that differences in the matS sequences do not modify MatP's ability to bind to the chromosome and affect its capacity to inhibit MukBEF, these results suggested that the density of matS sites in a small chromosomal region has a greater impact than dispersion of the same number of matS sites over a larger segment”

Figure 5: show selected examples of matS site distribution in addition to the averaged distribution (as in supplemental figure)?

Figure 5 shows the median of the matS distribution based on the matS positions of 16 species as displayed in the supplementary figure. We believe that this figure is interesting as it represents the overall matS distribution across the Enterobacterales, Pasteurellales, and Vibrionales.

How do authors define 'background levels' (page 9)in their ChIP-Seq experiments? Please add a definition or reword.

We agree that the term 'background level' here could be confusing, so we have modified it to 'basal level' to refer to the non-replicating sequence. The background level can be observed in Supplementary Figure 9 in the ChIP without tags, and, on average, the background level is 1 throughout the entire chromosome in these control experiments.

This reviewer would naively expect the normalized ChIP-Seq signals to revolve around a ratio of 1 (Fig. 4)? They do in one panel (Figure 4B) but not in the others (Figure 4A). Please provide an explanation.

We thank the referee for this pertinent observation. An error was made during the smoothing of the data in Figure 4A, which resulted in an underestimation of the input values. This mistake does not alter the profile of the ChIP (it's a division by a constant) and our conclusions. We provide a revised version of the figure.

Inconsistent axis labelling: e.g Figure 4Enterobacterals should be Enterobacterales (?)KB should be kbMB should be MbImput should be InputFlaG should be Flag

We have made the suggested modifications to the text.

'These results unveiled that fluorescent MukBEF foci previously observed associated with the Ori region were probably not bound to DNA' Isn't the alternative scenario that MukBEF bound to distant DNA segments colocalize an equally likely scenario? Please rephrase.

Since we lack evidence regarding what triggers the formation of a unique MukB focus associated with the origin and what this focus could represent, we have removed this sentence.

**Reviewer #3 (Recommendations For The Authors):**
The text is well-written and easy to follow, but I would suggest several improvements to make things clearer:1. Many plots are missing labels or legends. (I) All contact plots such as Fig. 1C should have a color legend. It is not clear how large the signal is and whether the plots are on the same scale. (II)

As indicated in the materials and methods section, the ratio presented on this manuscript was calculated for each point on the map by dividing the number of contacts in one condition by the number of contacts in the other condition. The Log2 of the ratio was then plotted using a Gaussian filter.

1. Genotypes and strain names are often inconsistent. Sometimes ΔmukB, ΔmatP, ΔmatS is used, other times it is just mukB, matP, matS; There are various permutations of LiTer, Liter, liter etc.

These inconsistencies have been corrected.

1. The time notation is unconventional. I recommend using 0 min, 40 min, 120 min etc. instead of T0, T40mn, T2H.

As requested, we have standardized and used conventional annotations.

1. A supplemental strain table listing detailed genotypes would be helpful.

A strain table has been added, along with a second table recapitulating the positions of matS in the different strains.

1. Fig. 1A: Move the IPTG labels to the top? It took me a while to spot them.

We have moved the labels to the top of the figure and increased the font size to make them more visible.

1. Fig 1C: Have these plots been contrast adjusted? If so, this should be indicated. The background looks very white and the transitions from diagonal to background look quite sharp.

No, these matrices haven't been contrast-adjusted. They were created in MATLAB, then exported as TIFF files and directly incorporated into the figure. Nevertheless, we noticed that the color code of the matrix in Figure 3 was different and subsequently adjusted it to achieve uniformity across all matrices.

7, Fig 1C: What is the region around 3 Mb and 4 Mb? It looks like the contacts there are somewhat MukBEF-independent.

The referee is right. In the presence of the plasmid pPSV38 (carrying the MukBEF operon or not), we repeatedly observed an increase of long range contacts around 3 Mb. The origin of these contacts is unknown.

1. Fig 1D: Have the log ratios been clipped at -1 and 1 or was some smoothing filter applied? I would expect the division of small and noisy numbers in the background region to produce many extreme values. This does not appear to be the case.

The referee is right, dividing two matrices generates a ratio with extreme values. To avoid this, the Log2 of the ratio is plotted with a Gaussian filter, as described before (Lioy et al., 2018).

1. Fig 1E: I recommend including a wild-type reference trace as a point of reference.

We have added the WT profile to the figure.

1. Fig 2: I feel the side-by-side cartoon from Supplemental Fig. 2A could be included in the main figure to make things easier to grasp.

We added a schematic representation of the chromosome configuration on top of the matrices to aid understanding.

1. Fig. 2C: One could put both plots on the same y-axis scale to make them comparable.

We have modified the axes as required.

1. Fig. 3C: The LiTer4 ratio plot has two blue bands in the 3-4.5 Mb region. I was wondering what they might be. These long-range contacts seem to be transposition-dependent and suppressed by MatP, is that correct?

The referee is right. This indicates that in the absence of MatP, one part of the Ter was able to interact with a distal region of the chromosome, albeit with a low frequency. The origin is not yet known.

1. Fig. 3E: It is hard to understand what is a strain label and what is the analyzed region of interest.The plot heading and figure legend say Ter2 (but then, there are different Ter2 variants), some labels say Ter, others say Ter2, sometimes it doesn't say anything, some labels say ΔmatS or ΔmatP, others say matS or matP, and so on.

We have unified our notation and add more description on the legend to clarify this figure :

“Ter” corresponds to the range of contacts over the entire Ter region, in the WT strain (WT Ter) or in the ΔmatP strain (ΔmatP Ter). The column WT matSX-Y corresponds to the range of contacts between the designated matS sites in the WT configuration. This portion of the Ter can be compared with the same Ter segment in the transposed strain (Ter2). Additionally, the matS20-28 segment corresponds to Ter2 in LiTer9, just as matS22-28 corresponds to Ter2 in LiTer7, and matS25-28 to Ter2 in LiTer4. The range of contacts of this segment was also measured in a ΔmatP or ΔmatS background.”

1. Fig. 4 and p.9: "Normalized ChIP-seq experiments were performed by normalizing the quantity of immuno-precipitated fragments to the input of MukB-Flag and then divide by the normalized ChIP signals at t0 to measure the enrichment trigger by replication."This statement and the ChIP plots in Fig. 4A are somewhat puzzling. If the data were divided by the ChIP signal at t0, as stated in the text, then I would expect the first plot (t0) to be a flat line at value 1. This is not the case. I assume that normalized ChIP is shown without the division by t0, as stated in the figure legend.

The referee is right. This sentence has been corrected, and as described in the Methods section, Figure 4 shows the ChIP normalized by the input.

If that's true and the numbers were obtained by dividing read-count adjusted immunoprecipitate by read-count adjusted input, then I would expect an average value of 1. This is also not the case. Why are the numbers so low? I think this needs some more details on how the data was prepared.

The referee is right; we thank him for this remark. Our data are processed using the following method: the value of each read is divided by the total number of reads. A sliding window of 50 kb is applied to these normalized values to smooth the data. Then, the resulting signal from the ChIP is divided by the resulting signal from the input. This is what is shown in Figure 4. Unfortunately, for some of our results, the sliding window was not correctly applied to the input data. This did not alter the ChIP profile but did affect the absolute values. We have resolved this issue and corrected the figure.

Another potential issue is that it's not clear what the background signal is and whether it is evenly distributed. The effect size is rather small. Negative controls (untagged MukB for each timepoint) would help to estimate the background distribution, and calibrator DNA could be used to estimate the signal-to-background ratio. There is the danger that the apparent enrichment of replicated DNA is due to increased "stickiness" rather than increased MukBEF binding. If any controls are available, I would strongly suggest to show them.

To address this remark, a ChIP experiment with a non-tagged strain under comparable synchronization conditions has been performed. The results are presented as Supplementary Figure 9; they reveal that the enrichment shown in Figure 4 is not attributed to nonspecific antibody binding or 'stickiness’.

1. Fig. 4A, B: The y-axes on the right are unlabeled and the figure legends mention immunoblot analysis, which is not shown.

We labeled the y-axes as 'anti-Flag ChIP/input' and made corrections to the figure legend.

1. Fig. 4B: This figure shows a dip in enrichment at the Ter2 region of LiTer7, which supports the authors' case. Having a side-by-side comparison with WT at 60 min would be good, as this time point is not shown in Fig. 4A.

Cell synchronization can be somewhat challenging, and we have observed that the timing of replication restart can vary depending on the genetic background of the cells. This delay is evident in the case of LiTer7. To address this, we compared LiTer7 after 60 minutes to the wild type strain (WT) after 40 minutes of replication. Even though the duration of replication is 20 minutes longer in LiTer7, the replication profiles of these two strains under these two different conditions (40 minutes and 60 minutes) are comparable and provide a better representation of similar replication progression.

1. Fig. 4C: Highlighting the position of the replication origin would help to interpret the data.

We highlight oriC position with a red dash line

1. Fig. 4C: One could include a range-of-contact plot that compares the three conditions (similar to Fig. 1E).

We have added this quantification to Supplemental Figure 8

1. Supplemental Fig. 2A: In the LiTer15 cartoon, the flanking attachment sites do not line up. Is this correct? I would also recommend indicating the direction of the Ter1 and Ter2 regions before and after recombination.

In this configuration, attB and attR, as well as attL and attB', should be aligned but the remaining attR attL may not. We have corrected this misalignment.To clarify the question of sequence orientation, we have included in the figure legend that all transposed sequences maintain their original orientation.

1. Supplemental Fig. 3: One could show where the deleted matS sites are.

We added red asterisks to the ChIP representation to highlight the positions of the missing matS.

1. Supplemental Fig. 3B: The plot legend is inconsistent with panel A (What is "WT2")?

We have corrected it.

1. Supplemental Fig. 3C: The E-value notation is unusual. Is this 8.9 x 10^-61?

The value is 8.9 x 10-61; we modified the annotation.

1. Abstract: "While different features for the activity of the bacterial canonical SMC complex, SmcScpAB, have been described in different bacteria, not much is known about the way chromosomes in enterobacteria interact with their SMC complex, MukBEF."

Could this be more specific? What features are addressed in this manuscript that have been described for Smc-ScpAB but not MukBEF? Alternatively, one could summarize what MukBEF does to capture the interest of readers unfamiliar with the topic.

We modified these first sentences.

1. p.5 "was cloned onto a medium-copy number plasmid under control of a lacI promoter" Is "lacI promoter" correct? My understanding is that the promoter of the lacI gene is constitutive, whereas the promoter of the downstream lac operon is regulated by LacI. I would recommend providing an annotated plasmid sequence in supplemental material to make things clearer.

We modified it and replaced “ lacI promoter” with the correct annotation, pLac.

1. p. 5 heading "MukBEF activity does not initiate at a single locus" and p. 6 "Altogether, the results indicate that the increase in contact does not originate from a specific position on the chromosome but rather appears from numerous sites". Although this conclusion is supported by the follow-up experiments, I felt it is perhaps a bit too strong at this point in the text. Perhaps MukBEF loads slowly at a single site, but then moves away quickly? Would that not also lead to a flat increase in the contact plots? One could consider softening these statements (at least in the section header), and then be more confident later on.

We used 'indicate' and 'suggesting' at the end of this results section, and we feel that we have not overreached in our conclusions at this point. While it's true that we can consider other hypotheses, we believe that, at this stage, our suggestion that MukBEF is loaded over the entire chromosome is the simplest and more likely explanation.

1. p.7: "[these results] also reveal that MukBEF does not translocate from the Ori region to the terminus of the chromosome as observed with Smc-ScpAB in different bacteria."This isn't strictly true for single molecules, is it? Some molecules might translocate from Ori to Ter. Perhaps clarify that this is about the bulk flux of MukBEF?

At this point, our conclusion that MukBEF does not travel from the ori to Ter is global and refers to the results described in this section. However, the referee is correct in pointing out that we cannot exclude the possibility that in a WT configuration (without a Ter in the middle of the right replicore), a specific MukBEF complex can be loaded near Ori and travel all along the chromosome until the Ter. To clarify our statement, we have revised it to 'reveal that MukBEF does not globally translocate from the Ori region to the terminus of the chromosome.' This change is intended to highlight the fact that we are drawing a general conclusion about the behavior of MukBEF and to facilitate its comparison with Smc-ScpAB in *B. subtilis*.

1. p. 10: The section title "Long-range contacts correlate with MukBEF binding" and the concluding sentence "Altogether, these results indicate that MukBEF promotes long-range DNA contacts independently of the replication process even though it binds preferentially in newly replicated regions" seem to contradict each other. I would rephrase the title as "MukBEF promotes long-range contacts in the absence of replication" or similar.

We agree with this suggestion and have used the proposed title.

1. p. 13: I recommend reserving the name "condensin" for the eukaryotic condensin complex and using "MukBEF" throughout.

We used MukBEF throughout.